# Adjoint-Based Simultaneous State and Parameter Estimation in an Arctic Sea Ice-Ocean Model using MITgcm (c63m)

Guokun Lyu[12], Longjiang Mu[3], Armin Koehl[4] , Ruibo Lei[1*], Xi Liang[5], and Chuanyu Liu[6]

[1]Key Laboratory for Polar Science, Polar Research Institute of China, Ministry of Natural Resources, Shanghai, 200136, China
[2]Shanghai Key Laboratory of Polar Life and Environment Sciences, School of Oceanography, Shanghai Jiao Tong University, Shanghai, 200230, China
[3]Laoshan Laboratory, Qingdao, 266100, China
[4]Center for Earth System Research and Sustainability (CEN), Universit ä Hamburg, Hamburg, 54662, Germany
[5]Key Laboratory of Marine Hazards Forecasting, National Marine Environmental Forecasting Center, Ministry of Natural Resources, Beijing, 100081,China
[6]Key Laboratory of Ocean Observation and Forecasting, and Key Laboratory of Ocean Circulation and Waves, Institute of Oceanology, Chinese Academy of Sciences, Qingdao, 266000, China

*Corresponding to*: Ruibo Lei (leiruibo@pric.org.cn)

**Abstract.** Parameters in sea ice-ocean coupled models greatly affect the simulated evolution of the ocean and sea ice, and are typically tuned to bring the model state close to observations. Using an adjoint method, spatiotemporally varying parameters of an Arctic sea ice-ocean coupled model are optimized simultaneously with the initial conditions and atmospheric forcing by assimilating satellite and in-situ observations. The assimilation results show that the joint state and parameter estimation (SPE) substantially improves the sea ice concentration simulations. Particularly in October, when the ocean surface starts to refreeze, SPE reduces the lead closing parameter $H_o$ (which determines the minimum ice thickness formed in the open water), thereby increasing the sea ice growth and facilitating the seasonal rapid sea ice recovery in the Arctic's Pacific sector. Comparisons with sea ice thickness observations from the moored upward-looking sonars and Ice Mass Balance buoys demonstrate that incorporating optimized model parameters into the coupled model also leads to better sea ice thickness estimation. Given that the optimal set of sea ice parameters may evolve alongside the thinning of Arctic sea ice, the adjoint-based SPE scheme has the potential to more accurately reconstruct the histical Arctic ocean and sea ice changes covering the satellite era, supporting research on Arctic sea ice and ocean variability.

## 1 Introduction

In climate models, parameterizations are widely used to simulate the effects of unresolved processes on large-scale state variables. The parameters in these parameterizations significantly affect model simulations across various spatial and temporal scales (e.g., Mauritsen et al., 2012; Murphy et al., 2004). This is particularly important for the Arctic system, where observations are extremely scarce, and the ocean and sea ice states are undergoing rapid changes.

Most model parameters cannot be measured directly. Usually, a globally uniform value for individual parameters is assumed as the default in parameterization schemes. When incorporated into complex climate models, sensitive parameters are usually further adjusted to bring the model's climatology closer to observed state (e.g., Hourdin et al., 2017; Mauritsen et al., 2012). The underlying assumption is that these sensitive parameters determine the model's climatology and variations. Over the past decades, manual (e.g., Mauritsen et al., 2012) and automatic tuning (e.g., Jackson et al., 2008; Williamson et al., 2013; Yang et al., 2013) methods have been developed to optimize a number of sensitive parameters (on the order of 10). In addition, data assimilation techniques, including ensemble Kalman filters (Annan et al., 2005; Massonnet et al., 2014; Wu et al., 2012) and the adjoint method (Liu et al., 2012; Lyu et al., 2018), have been tested for optimizing spatiotemporally varying parameters with large dimensions.

In sea ice models, parameters significantly affect the sea ice's seasonal and interannual evolutions, as well as the heat, freshwater, and momentum budgets of the sea ice-ocean-atmosphere system (e.g., Kim et al., 2006; Uotila et al., 2012; Nie et al., 2023). Similar to climate models, these parameters are manually tuned through a trial-and-error process to match the historical sea ice state (e.g., Hibler, 1979; Miller et al., 2005). Automated parameter estimation methods, such as genetic algorithms (Sumata et al., 2019), ensemble Kalman filters (Massonnet et al., 2014), and the adjoint method (Lu et al., 2021; Lyu et al., 2018; Panteleev et al., 2020), are applied to optimize sensitive parameters using satellite sea ice observations. More recently, machine learning (e.g., Bretherton et al., 2022; Horvat & Roach, 2022; Kochkov et al., 2024) has been applied to develop the sub-grid parameterizations using high-resolution model simulations and reanalysis datasets. However, improvements in observed variables often come with detrimental effects on the other variables or in some regions. For instance, Massonnet et al. (2014) reported that the improvements in sea ice speed were accompanied by an overestimation of winter ice areal outflow through the Fram Strait.

Several factors are likely responsible for these detrimental effects. First, model parameters represent only parts of the model's uncertain inputs, and they may be over-adjusted to compensate for errors in the other uncertain inputs. For instance, Miller et al. (2005) suggested that optimal parameters likely compensate for deficiencies in atmospheric forcing. Therefore, it is necessary to optimize the model's uncertain inputs simultaneously. Second, the sensitivities of the sea ice state to parameter perturbations show distinct spatiotemporal patterns (Kim et al., 2006; Miller et al., 2005), suggesting that allowing optimized parameters to vary geographically and temporally is likely to further improve model performance. Additionally, it is necessary to use more comprehensive Arctic ocean and sea ice observations to constrain model parameters and the other uncertain inputs.

In the framework of data assimilation, simultaneous state and parameter estimation (SPE) can be achieved by including spatiotemporally varying parameters in the set of control variables (e.g., Hu et al., 2010; Liu et al., 2012; Wu et al., 2012; Zhang, 2011). Since initial conditions typically exert significant influences on short timescales within the Earth system, while parameters are more important on longer timescales (e.g., Branstator & Teng, 2010),

constructing an error propagation model to project model-data misfits onto both initial conditions and parameters is
challenging. Using an ensemble Kalman filter with a short assimilation window (within the predictability limit of the
chaotic system), Zhang (2011) proposed to optimize the initial state first until the model-data misfits reach a quasi-
equilibrium, where the parameter errors dominate; then they optimized the model parameters to reduce residual errors.
Liu et al. (2012) used an adjoint model to project the model-data misfits over a large assimilation window onto initial
conditions, atmospheric forcing, and eddy mixing parameters.
Applications of SPE in ocean models (Liu et al., 2012) and coupled climate models of intermediate complexity
(Wu et al., 2012) have demonstrated additional effects on mitigating model bias and improving prediction accuracy.
However, to date, parameter estimation (Lu et al., 2021; Panteleev et al., 2020) and state estimation (e.g., Fenty et al.,
2017;Forget et al., 2015; Lindsay & Zhang, 2006; Lyu et al., 2021b; Mu et al., 2018; Nguyen et al., 2021; Liu et al.,
2018) are performed separately in the sea ice-ocean coupled model. Given the rapid thinning of the Arctic sea ice and
its extremely strong spatial heterogeneity, it is necessary to have parameters that vary spatiotemporally in response to
the changing climate background. Therefore, we developed SPE in a sea ice-ocean coupled model and investigated its
potential effects by assimilating satellite and in-situ measurements.
This paper is organized as follows. In Section 2, we introduce the coupled sea ice-ocean assimilation system,
including the model configuration, the parameters being adjusted, as well as the assimilated and independent
observations. Section 3 briefly evaluates the parameter sensitivities and accuracy of the adjoint model based on the
tangent linear model. Section 4 discusses the added effects of SPE. Finally, we summarize this study in Section 5.

## 2 Methodology

### 2.1 The Pan-Arctic Ocean and Sea Ice Assimilation System

The sea ice-ocean coupled assimilation system is based on the adjoint method (4D-Var in operational oceanic
and meteorological forecasting communities) from the Estimating the Circulation and Climate of the Ocean (ECCO)
project. It employs the Massachusetts Institute of Technology general circulation model (MITgcm, Marshall et al.,
1997) coupled with a zero-layer dynamic and thermodynamic sea ice model (Hibler, 1979; Hibler, 1980; Losch et al.,
2010). The sea ice dynamics are based on a viscous-plastic rheology and solved using a line successive over-relaxation
algorithm (Zhang & Hibler, 1997).
The adjoint of the sea ice-ocean coupled model is generated using the Transformation of Algorithms in Fortran
(Giering & Kaminski, 1998; Heimbach et al., 2010). In previous studies, due to persistent instability issues, the adjoint
of the sea ice dynamics was usually excluded (Fenty & Heimbach, 2013; Fenty et al., 2017; Nguyen et al., 2021) or
simplified to the adjoint of free-drift sea ice dynamics (Koldunov et al., 2017; Lyu et al., 2021a; Lyu et al., 2021b) in
assimilation experiments. Recently, Lyu et al. (2023) incorporated and approximated the adjoint of viscous-plastic sea
ice dynamics. In this study, we use the same model configuration as Lyu et al. (2023). Horizontally, we use a
curvilinear grid with a resolution of 12−18 km. Vertically, we have 50 z-levels with layer thickness ranging from 10
m at the surface to 456 m in the deep ocean. In addition, we further developed the adjoint model to jointly estimate
the state and spatiotemporally varying parameters, and evaluated the added effects by comparing with the results in
Lyu et al. (2023).
The adjoint method minimizes a quadratic function ($J$) iteratively by adjusting the control variables ($C$) to bring
the model simulation closer to available observations:
$$J(C) = \sum_{t=1}^{T1}[y(t) - E(t)x(t)]^T R^{-2}[y(t) - E(t)x(t)] + \Delta C B^{-2} \Delta C^T \tag{1}$$
The first term on the right-hand side computes the "distance" between observations ($x$) and corresponding model-
simulated variables ($y$) over a long period (one year in this study). The model-data misfits are normalized by their
prior error ($R$). $E$ interpolates the model state to the observations. Values of $R$ for different observation types are
introduced in Section 2.2.
The second term on the right-hand side measures the magnitude of the adjustments to the control variables ($\Delta C$)
and $B^{-2}$ is the background error covariance. In the 3D-Var and 4D-Var system with short assimilation windows (on
the order of days), covariance matrix $B^{-2}$ is critical for filling in observational gaps and achieving the multivariate
adjustments to initial conditions (Li et al., 2008; Weaver & Mirouze, 2013). However, on longer timescales (several
months to years), this is no longer necessary because the advection and diffusion in the adjoint model transport
information about model-data misfits to locations without observations and to unobserved variables. Therefore, for
simplicity, we use a diagonal $B$ matrix, and the second term on the right-hand side of equation (1) is simplified to $\frac{\Delta C^2}{\sigma^2}$,
where $\sigma$ represent the corresponding errors. As this term increases quadratically with $\Delta C$, it limits the magnitude of
adjustments during optimization.
In addition to the initial oceanic temperature and salinity, sea ice concentration (SIC), sea ice thickness (SIT),
and daily atmospheric state, which includes 2-m air temperature, 2-m specific humidity, precipitation rate, 10-m wind
vectors, downward longwave radiation, and net shortwave radiation, we further include 13 spatiotemporally varying
model parameters (see Table 1) in the control variables. These parameters vary by geographic location and on a daily
basis. The uncertainties of 13 parameters are not precisely known; their error distributions are assumed to be Gaussian,
with standard deviations of 20% of their default values.
The assimilation system iteratively minimizes the cost function $J$ using a quasi-Newton algorithm (Gilbert &
Lemaréchal, 2006) and gradient information $\frac{\partial J}{\partial c}$. The optimization stops when the cost function can no longer be
reduced. The adjoint of the ocean-sea ice coupled model is used to compute $\frac{\partial J}{\partial c}$ which has dimensions of $10^8$. To ensure
the stability of the adjoint model over a one-year period, modifications to the adjoint codes are handled carefully.
Details are provided in Lyu et al. (2023). The accuracy of the adjoint model is presented in Section 3.
**2.2 The Assimilation Experiments and the Control Variables**
Two assimilation experiments and one control run were performed to examine the additional effects of SPE. The
initial condition for January 1, 2012, is taken from the Arctic sea ice-ocean reanalysis by Lyu et al. (2021b), which
used the same model configuration as in this study. We take the forward simulation at iteration 0 as the control run
(CTRL). The two assimilation experiments differ in their choice of control variables. In the first assimilation
experiment, only the initial conditions and atmospheric forcings are adjusted (hereinafter named opti-SE). In the
second assimilation experiment, we further include 13 spatiotemporally varying model parameters (see Table 1 for
details) in the control variables (hereinafter named opti-SPE). Based on a previous study with a similar sea ice module
(Sumata et al., 2019) and our sensitivity experiments, we identified these 13 sea ice parameters, which have
considerable impact on sea ice properties. To prevent over-adjustments of the parameters, which may result in model
instability, we further set upper and lower bounds for individual parameters in the model (Table 1).
Table 1. The sea ice model parameters applied as additional control variables for optimization

| Parameters | Description | Default Values | Lower/Upper bounds |
|---|---|---|---|
| $\alpha_{wice}$ | Albedo of melting ice | 0.66 | 0.50, 0.95 |
| $\alpha_{dice}$ | Ice albedo | 0.75 | 0.50, 0.95 |
| $\alpha_{wsnow}$, | Albedo of melting snow | 0.71 | 0.50, 0.95 |
| $\alpha_{dsnow}$ | Snow albedo | 0.83 | 0.50, 0.95 |
| $Cond_{ice}$ | Ice conductivity (W·m$^{-1}$·K$^{-1}$) | 2.17 | 1.50, 3.00 |
| $Cond_{snow}$ | Snow conductivity (W·m$^{-1}$·K$^{-1}$) | 0.31 | 0.15, 0.40 |
| $C_{d\_wind}$ | Wind-ice drag coefficient | $2.0\times10^{-3}$ | $1.0\times10^{-3}$, $5.0\times10^{-3}$ |
| $C_{d\_water}$ | Water-ice drag coefficient | $5.5\times10^{-3}$ | $1.0\times10^{-3}$, $10.0\times10^{-3}$ |
| $H_o$ | Lead closing parameter, which determines the minimum sea ice thickness formed in open water (m) | 0.5 | 0.03, 1.0 |
| $P*$ | Ice compressive strength constant (N m$^{-2}$) | $2.8\times10^{4}$ | $1.0\times10^{4}$, $5.0\times10^{4}$ |
| $C*$ | Ice strength decay constant | -20.0 | -30.0, -5.0 |
| $Es$ | Eccentricity of the yield curve describing the viscous-plastic rheology | 2.0 | 1.5, 2.3 |
| $F_T$ | Friction velocity for ice-ocean heat flux (m s$^{-1}$) | $0.88\times10^{-3}$ | $0.4\times10^{-3}$, $1.2\times10^{-3}$ |


## 2.2 Assimilated Observations

We assimilate both satellite and in-situ observations (listed in Table 2). In the assimilation process, the
observations and their errors must be provided beforehand. Errors for these types of observations are described in
previous studies (Lyu et al., 2023).
Altimeter along-track L3 sea surface heights anomalies (SLA) are assimilated with observational errors of 3 cm.
Gridded sea surface temperature (SST) data for open waters and their associated errors are based on optimally
interpolated microwave and infrared data from the Remote Sensing System (Gentemann et al., 2004). Subsurface
oceanic temperature and salinity data are based on the EN4 dataset (Good et al., 2013). Observational errors of
temperature and salinity (T&S) profiles are depth-dependent, ranging from ~0.6 °C and ~0.3 PSU at the surface to
~0.02 °C and ~0.02 PSU in the deep ocean. The World Ocean Atlas 2018 (WOA18) climatology is also assimilated
to reduce biases in ocean temperature and salinity, with their prior errors set to five times those of the in-situ profiles.
Since we interpolate the WOA18 to the finer model grids, which implicitly increases the number of the observations,
we reduce the weight of temperature and salinity climatology cost components by multiplying by a factor of 0.04.
We assimilate the SIC observations from the Special Sensor Microwave/Imager (SSM/I) and ARTIST Sea Ice
(ASI) algorithm (Kaleschke et al., 2001; Spreen et al., 2008). This dataset does not provide observational error
estimates. In data assimilation studies, the observational errors should consist of representation and instrumental errors.
Considering larger SIC errors near coastlines due to reduced accuracy in the SIC product and poor representation of
landfast ice in the model (Fenty & Heimbach, 2013), we set the representation errors to 15% within 50 km of the
coastline and 10% in other areas. Additionally, considering the dependence of SIC errors on the absolute value of SIC
(Chen et al., 2023), we multiply the representation errors by factors of 0.85, 1.20, 1.10, and 1.00 for observed SIC
ranges of $0\%, < 15\%, 15\% - 25\%$, and $> 25\%$, respectively.

Satellite SIT observations are based on the merged CryoSat-2 and SMOS product (CS2SMOS), which includes

observational error estimates (Ricker et al., 2017). The SIT in this dataset represents the mean SIT over 25 km×25 km
areas but excludes open waters. Note that since the model simulates the grid-mean SIT (SIT×SIC), we convert the
model-simulated grid-mean SIT to the mean SIT over ice-covered regions before comparing it against the satellite
SIT observations in the cost function. Satellite sea ice drift (SID) data are derived from the Ocean & Sea Ice Satellite
Application Facility (OSISAF, https://osi-saf.eumetsat.int) product at a resolution of 62.5 km, which applies a
continuous maximum cross-correlation method to track the sea ice displacements over a 48-hour period from
sequences of satellite images (Lavergne et al., 2010). The errors are dominated by satellite resolution (10–15 km), and
are set to 10 km every 2 days (~0.06 m s$^{-1}$).
Table 2. Assimilated Observations.

| Observations[1] | Resolution | Number | Source | Reference |
|---|---|---|---|---|
| | | | | |
| SLA | 7.0 km | $7.6×10^5$ | http://marine.copernicus.eu | (Pujol et al., 2016; Taburet et al., 2019) |
| SST | 25.0 km | $2.0×10^7$ | https://data.remss.com/SST/daily/mw_ir/v05.1/netcdf/, (last access: 13 April 2025) | (Gentemann et al., 2004) |
| T&S | – | $5.0×10^5$ | https://www.metoffice.gov.uk/hadobs/en4/download-en4-2-2.html, (last access: 13 April 2025) | (Good et al., 2013) |
| SIC | 25.0 km | $3.6×10^7$ | https://thredds.met.no/thredds/catalog/osisaf/met.no/reprocessed/ice/conc_450a_files/catalog.html (last access: 13 April 2025) | (Lavergne et al.,2019) |
| SIT | 25.0 km | $8.9×10^6$ | https://spaces.awi.de/display/CS2SMOS/CryoSat-SMOS+Merged+Sea+Ice+Thickness (last access: 13 April 2025) | (Ricker et al., 2017) |
| SID | 62.5 km | $5.8×10^5$ | https://thredds.met.no/thredds/catalog/osisaf/met.no/ice/drift_lr/merged/catalog.html (last access: 13 April 2025) | (Lavergne et al., 2010) |
| WOA18 | 1° | $2.9×10^7$ | https://www.ncei.noaa.gov/access/world-ocean-atlas-2018/ (last access: 13 April 2025) | (Locarnini et al., 2018; Zweng et al., 2018) |

[1] SLA sea surface height anomaly, SST sea surface temperature, T&S temperature and salinity profiles from EN4
datasets, SIC sea ice concentration, SIT sea ice thickness, SID sea ice drift, WOA18 World Ocean Atlas 2018
**2.3 Independent Observations**

Independent sea ice observations are used to validate the assimilation results. The observations can be categorized

into two groups: 1) single-point measurements, which include sea ice variables within a small area or at specific sites,
such as SIT or snow depth from the Ice Mass Balance buoys (IMBs) and ice draft from moored upward-looking sonars
(ULSs); 2) gridded measurements, such as remote sensing products. These two types of measurements are not fully
consistent, especially in regions with strong spatial heterogeneity of sea ice and prevalent ice dynamic processes.
Nevertheless, we use these data to validate and cross-validate the assimilation results and gridded SIT observations,
identifying potential discrepancies among the assimilated datasets, the numerical model, and the point observations.
**2.3.1 Ice Mass Balance Buoy**
IMBs are ice-based observing systems that measure the evolution of snow and sea ice growth and ablation, as
well as vertical temperature profiles through the snow, ice, and upper ocean along ice drift trajectories. It uses two
acoustic sensors to measure the ice surface and bottom. SIT is derived from the distance between ice surface and
bottom. The errors in measurements of the ice surface and bottom by each acoustic sounders are ± 5mm (Richter-
Menge et al., 2006). IMBs are deployed on thick and level ice floes to achieve relatively long measurement periods.
It must be acknowledged that directly comparing this point measurements of SIT with gridded SIT is not
straightforward. Thus, we also compare the mean SIT bias and temporal changes in SIT along the drifting trajectories.
Compared to the satellite or helicopter-based altimeter observations, the IMB observations can effectively characterize
the thermodynamic thickening of sea ice (Koo et al., 2021; von Albedyll et al., 2022).
**2.3.2 International Arctic Buoy Programme**
The International Arctic Buoy Programme (IABP, https://iabp.apl.uw.edu/index.html) collects ice-based drifting
buoys from various organizations. Most of these buoys also monitor sea level pressure and surface air temperature
along the ice drift trajectories. In this study, we compute sea ice velocities based on drift trajectories and compare
them with the model simulations. For 2012, we selected 23 trajectories with continuous position observations of more
than 100 days for model validation.
**2.3.3 Mooring measurements from the Beaufort Gyre Exploration Project**
Starting from 2003, the Beaufort Gyre Exploration Project (BGEP), based at the Woods Hole Oceanographic
Institution (https://www2.whoi.edu/site/beaufortgyre/), deployed ULSs at three locations: $M_a$, $M_b$, and $M_d$ (see Figure
7). The ULSs measure local sea ice draft with an error of ± 0.1m (Krishfield et al., 2014). In this study, we use daily-
averaged sea ice draft for comparisons. Sea ice drafts are converted to sea ice thickness by multiplying with a factor
of 1.1, calculated as ratio between mean seawater density (1024.0 kg m$^{-3}$) and sea ice density (910.0 kg m$^{-3}$), assuming
no snow on the sea ice (Nguyen et al., 2011).
**3 Parameter Sensitivities and Accuracy of the Tangent Linear Approximation**
There are two requirements for optimizing model parameters using observations. First, parameter changes within
the range of uncertainties have considerable impacts on model simulations. For instance, the SIC changes caused by
parameter uncertainties should be larger than observational errors. Second, the error propagation model (the tangent
linear model in this study) must accurately reproduce the spatiotemporal error propagations induced by the uncertain
model inputs.
Here, we adjust each of the 13 model parameters (see Table 1) individually by adding 10% of their default values
and integrate the model over 2012. We then evaluate the spatiotemporal changes in SIC normalized by satellite SIC
observational errors (e.g., 15%). Normalized SIC changes > 1.0 indicate that the SIC changes caused by 10%
parameter perturbations are detectable by satellite observations. As Figure 1 shows, albedos of the dry/wet snow and
ice ($\alpha_{dsnow}$, $\alpha_{dice}$, $\alpha_{wsnow}$, $\alpha_{wice}$,) have pronounced impacts on summertime SIC. The wind-ice drag coefficient ($Cd_{wind}$)
and ice compressive strength constant ($P^*$) also exhibit large impacts throughout the year. For the remaining
parameters, the normalized SIC changes grow slowly and show little seasonal dependence.

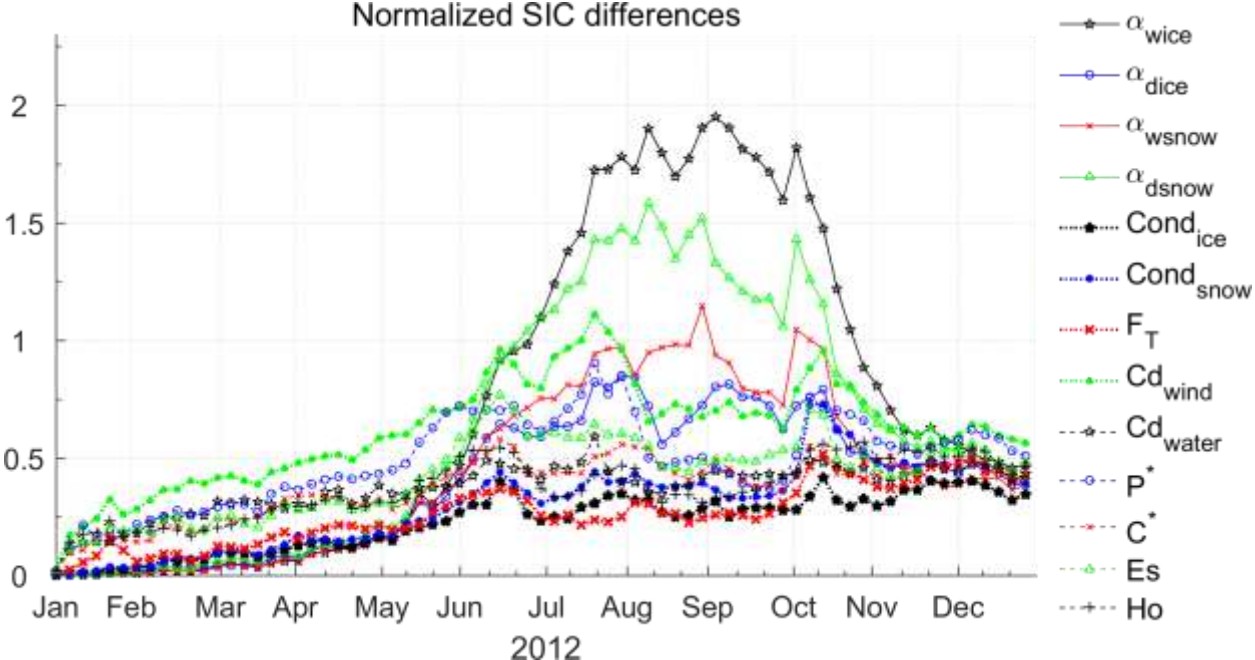


**Figure 1. Temporal evolutions of the norm of the SIC differences (normalized by 15% SIC) averaged over the ice-covered**
**regions due to the 10% perturbations on the 13 parameters (see the legend and Table 1 for description of the parameters).**

The accuracy of the tangent linear approximation is examined using the Taylor expansion. Taking the parameter
$H_o$ (lead closing parameter) as an example, we perturb it by $\Delta H_o = 10\% \times H_0$ and evaluate the first-order and higher-
order approximations as follows:
$$\Delta X_1 = M(H_o + \Delta H_o) - M(H_o) = \frac{\partial M}{\partial H} \cdot \Delta H_o + o(\Delta H_o{}^2) \tag{2},$$
$$\text{and } \Delta X_2 = M(H_o - \Delta H_o) - M(H_o) = -\frac{\partial M}{\partial H} \cdot \Delta H_o + o(-\Delta H_o{}^2) \tag{3}.$$
Here, $M$ is the model operator and $X$ represents the model variables. The linear (first-order) and nonlinear (higher-
order) errors can be obtained by $\frac{1}{2}(\Delta X_1 - \Delta X_2)$ and $\frac{1}{2}(\Delta X_1 + \Delta X_2)$. By integrating the tangent linear model with
$\Delta H_o = 10\% \times H_0$, we obtain error evolution of the model variables.
Although the overall impacts of $H_o$ on SIC do not exceed the SIC observational errors (normalized SIC changes
< 1.0; Figure 1) over the one-year period, it exerts substantial impacts on open water during refreezing (Figure 2a).
Since $H_o$ determines the initial sea ice thickness formed in open water, increasing $H_o$ by 10% will delay the formation
of new ice in open water, reducing sea ice coverage across the Arctic Ocean. The signals are dominated by the linear
component (Figure 2a), and the nonlinear component is small (Figure 2b). The tangent linear model can reproduce the
negative SIC changes very well (Figure 2c).

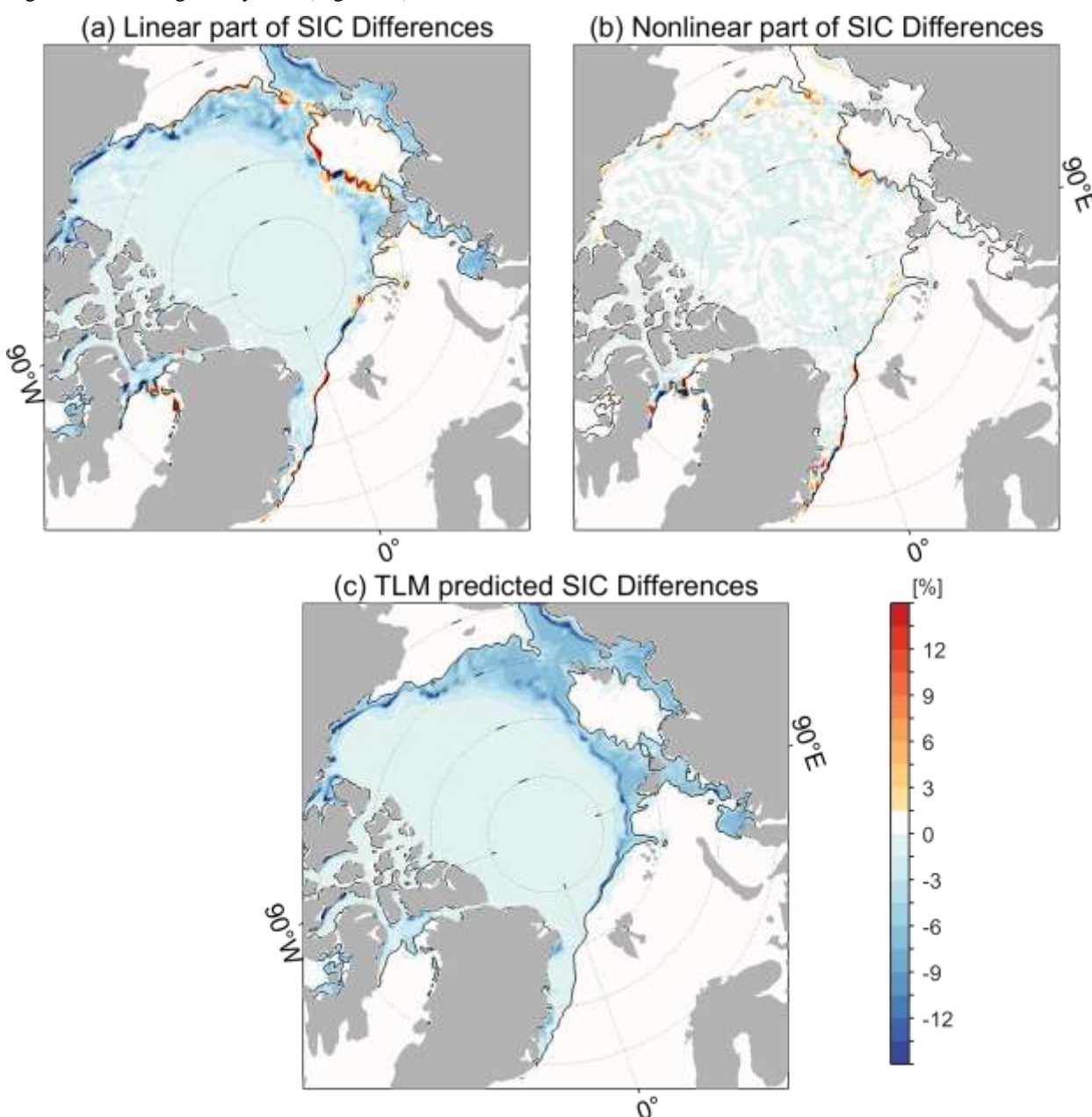


**Figure 2. (a) The linear and (b) nonlinear parts of SIC differences on November 11, 2012 computed using equations (2)-(3).**
**Panel (c) is the corresponding SIC changes predicted by the tangent linear model (TLM). The black lines denote the 15%**
**SIC contour.**

The above analysis demonstrates that the responses of SIC to parameter perturbations are dominated by the linear

component, and that the tangent linear model effectively captures the linear component of the error propagation,
reproducing both its pattern and amplitude. These results motivate us to further explore the potential of using
observations to simultaneously estimate the state and parameters.
**4 Evaluation of the Assimilation Results**
In this section, we evaluate the impacts of SPE on the estimated sea ice state. Additionally, we compare the two
assimilation runs and the control run against independent observations.
**4.1 The cost function reduction**
The optimization algorithm iteratively reduces the cost function $J$ using the gradients $\frac{\partial J}{\partial c}$. In the opti-SE and opti-
SPE, the optimizations stop at iterations 32 and 86, respectively. The cost functions are reduced to 59.8% and 49.3%
of their initial values (Figure 3), and the norms of the gradients ($\left|\frac{\partial J}{\partial c}\right|$) are reduced to 10.7% and 9.0%, respectively.
Of the assimilated observations, daily SST and SIC contribute to 27.3% and 42.9% of the cost function,
respectively, with these proportions reduced to 14.0% and 19.4% in opti-SE and to 10.1% and 13.2% in opti-SPE. For
other observations, their cost components are slightly reduced. In the following sections, we focus on improvements
in the SIC, SIT, and SID, and evaluate the simulations against independent observations.

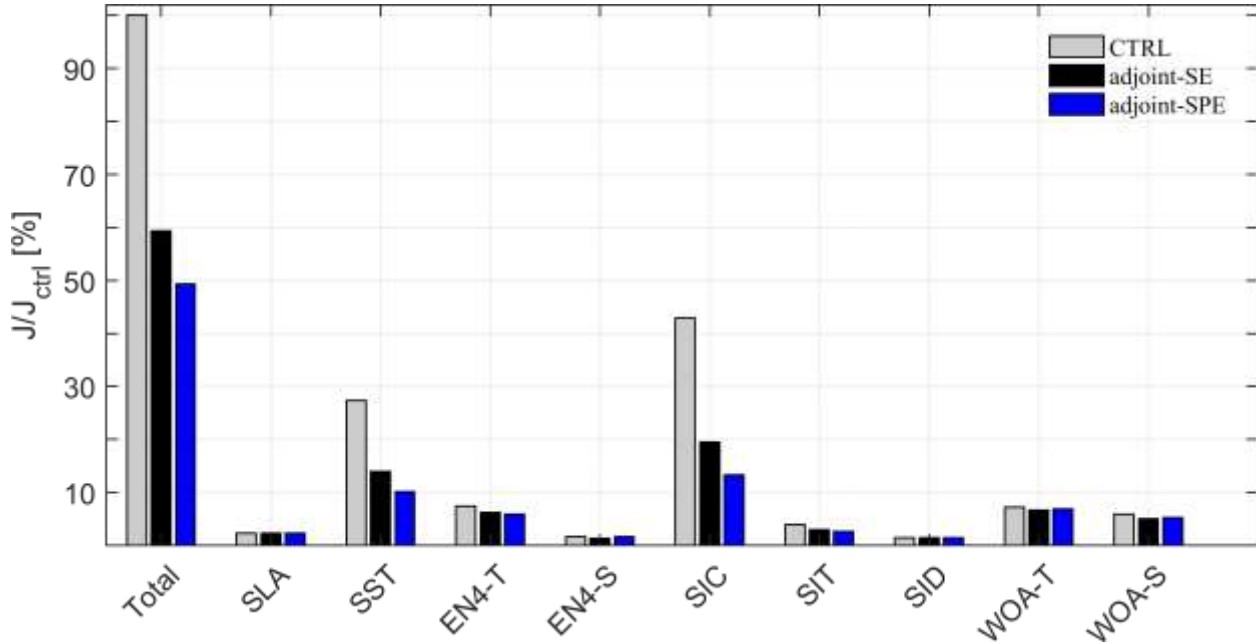

**Figure 3. The total cost function and individual components normalized by the total cost function in the control run ($J_{ctrl}$).**
**Abbreviations: SLA—sea level anomalies, SST—sea surface temperature, EN4-T—EN4 temperature profiles, EN4-**
**S—EN4 salinity profiles, SIC—sea ice concentration, SIT—sea ice thickness, SID—sea ice drift velocities; WOA—T and**
**WOA—S, climatological temperature and salinity from WOA18 dataset.**
**4.2 Improvements on Sea Ice Concentration**
In the control run (CTRL), normalized SIC errors grow rapidly during spring, peaking in June. They then decrease
from July to September before sharply increasing again in October (grey dotted line with filled circles in Figure 4).
opti-SE greatly improves the simulated SIC, with normalized SIC errors remaining near 1.0 throughout the year (black
line in Figure 4). However, the notable SIC errors persist from May to June, and the sharp October increase in errors
remain unmitigated. By jointly optimizing the model parameters, opti-SPE further reduces the SIC errors; in particular,
the errors from May to June and during October are reduced to acceptable levels (black line with filled circles in
Figure 4).

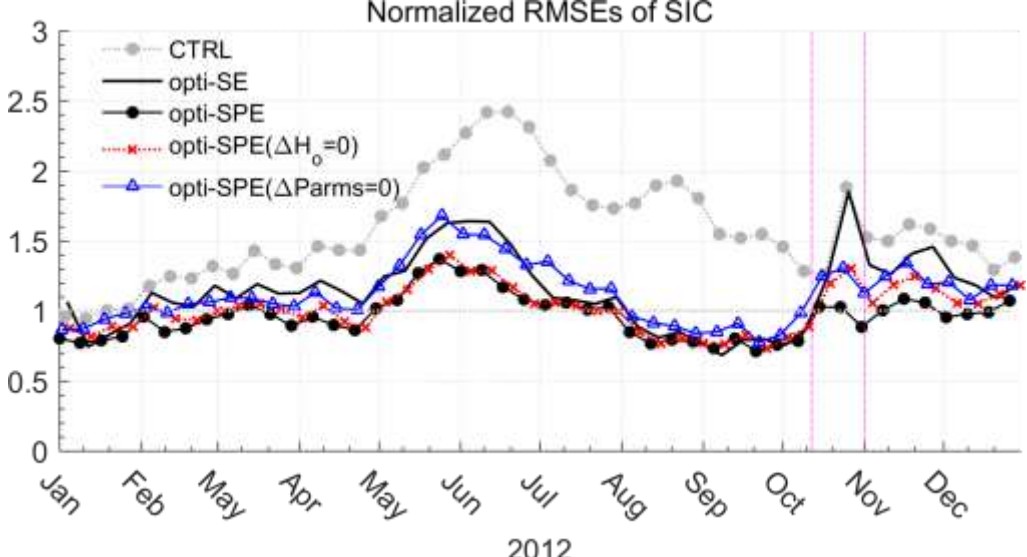

**Figure 4. Seasonal evolution of the root mean square errors of SIC (normalized by prior uncertainties) in CTRL, opti-SE,**
**and opti-SPE. In opti-SPE ($\Delta Ho=0$), we replace the optimized $Ho$ to its default value. And in opti-SPE ($\Delta Parms=0$), we reset**
**all the 13 parameters to their default values.**
Two additional simulations are presented in Figure 4 to demonstrate the added effects of joint parameter and state
estimation. By resetting all the 13 parameters to their default values (opti-SPE($\Delta Parms=0$), blue line with triangles in
Figure 4), SIC errors increase to a level similar to that in opti-SE (black line with filled circles in Figure 4). However,
in October, the SIC errors are smaller than opti-SE when reset the 13 parameters to their default values. This indicates
that the data assimilation finds a smaller minimum of the cost function $J$ which is inaccessable in opti-SE. The further
improvements in opti-SPE after October are partially attributed to adjustments to the parameter $H_o$, as evidenced by
the sharp increase in SIC errors when $\Delta H_o=0$ (opti-SPE ($\Delta H_o=0$), dotted red line with crosses in Figure 4).
Since the sea ice recovery process in October impacts ocean-sea ice-atmosphere fluxes and sea ice thickness in
subsequent years, we now focus on the SIC improvements during this period. Pronounced SIC differences between
the control run and satellite observations appear in the Arctic's Pacific sector (enclosed by the dashed black line in
Figure 5a–c), where the most extensive summer sea ice loss occurs across the Arctic Ocean. At the beginning of
October, the Arctic's Pacific sector is ice-free (red line with crosses in Figure 5d). The control run fails to reproduce
the ice-free conditions (grey line with filled circles in Figure 5d). Both opti-SE and opti-SPE reproduce the ice-free
conditions well (black line with trianges and blue line with rectangles in Figure 5d). Starting from October 10, the sea
ice begins to recover. Compared with the control run and observations, opti-SE underestimates SIC by more than 40%
(Figure 5b), and this underestimation persists until the early November (Figure 5d). By simultaneously adjusting the
model parameters, opti-SPE improves the sea ice recovery process and successfully reduces the negative SIC bias
(Figure 5c, d).

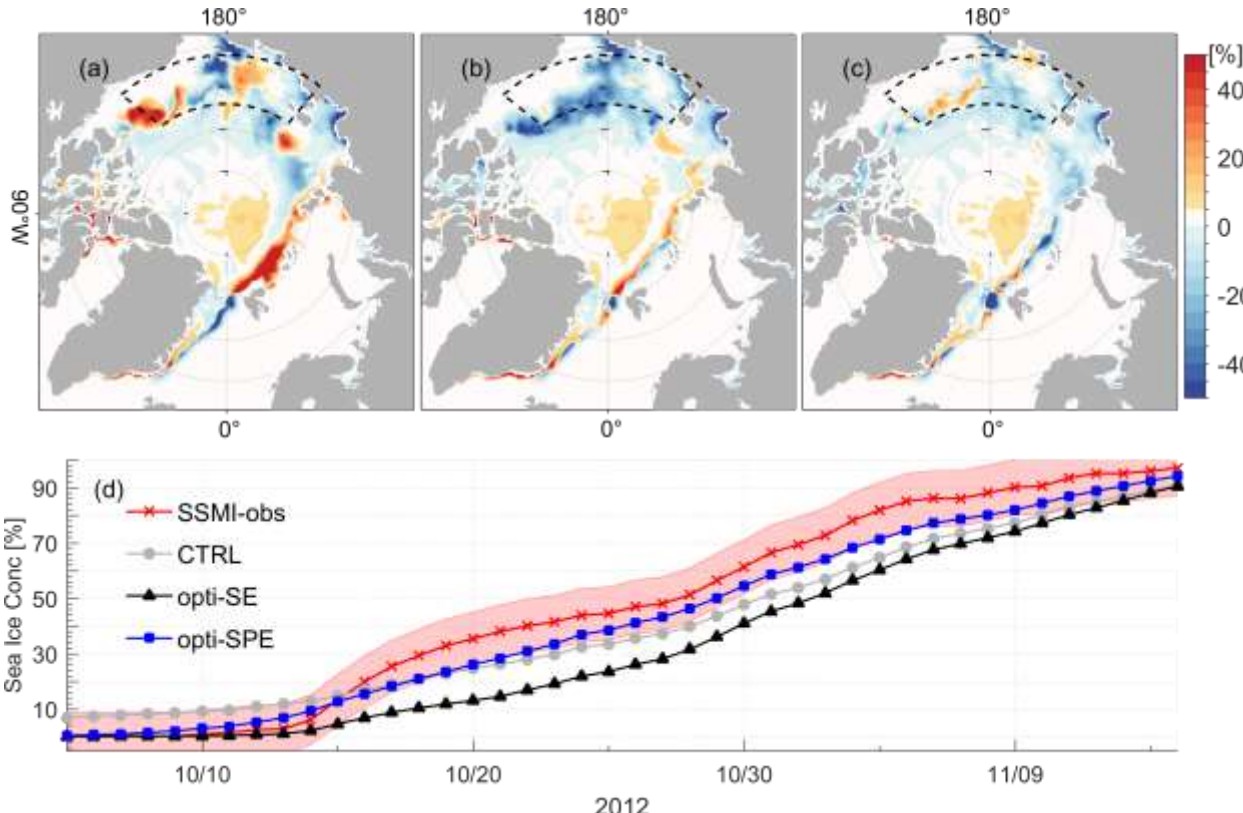

**Figure 5. Model-data differences of SIC averaged from October 12, 2012 to October 22, 2012 in (a) CTRL, (b) opti-SE,**
**and (c) opti-SPE. (d) Temporal evolution of in SIC averaged over the Pacific sector (enclosed by the dashed black line in**
**panels (a)–(c)) in the control run (CTRL), two assimilation runs (opti-SE and opti-SPE), and satellite observations (SSMI-**
**obs).**
During summer, open waters accumulate a large amount of heat, and this ocean heat must be released before the
ocean surface refreezes. Since the control run fails to reproduce the ice-free conditions (Figure 5d), more sea ice
persists in the Arctic's Pacific sector and less ocean heat is released before the open waters freeze. However, due to
the ice-free conditions in the summer season in opti-SE and opti-SPE, the ocean needs to release more heat before
refreezing. Therefore, opti-SE initially shows a much slower sea ice growth rate while opti-SPE improves the SIC
evolution by adjusting the parameter $H_o$. In the sea ice model, $H_o$ is the minimum SIT formed in open water, which
impacts the vertical and lateral growth of the sea ice and is set to 0.5 m by default. Over the ice-free water, decreasing
$H_o$ reduces the amount of latent heat that needs to be released during the initial formation process of sea ice. After
initial formation, the decreased $H_o$ accelerates the formation of thinner sea ice in open areas. In opti-SPE, $H_o$ is reduced
by more than 0.1 m over open water (Figure 6a) at the onset of refreezing (Figure 6b), which gives rise to the rapid
growth of thinner sea ice. The accelerated sea ice recovery significantly reduces the heat and momentum exchanges
between atmosphere and ocean, altering the feedback processes in the atmosphere-sea ice-ocean system.

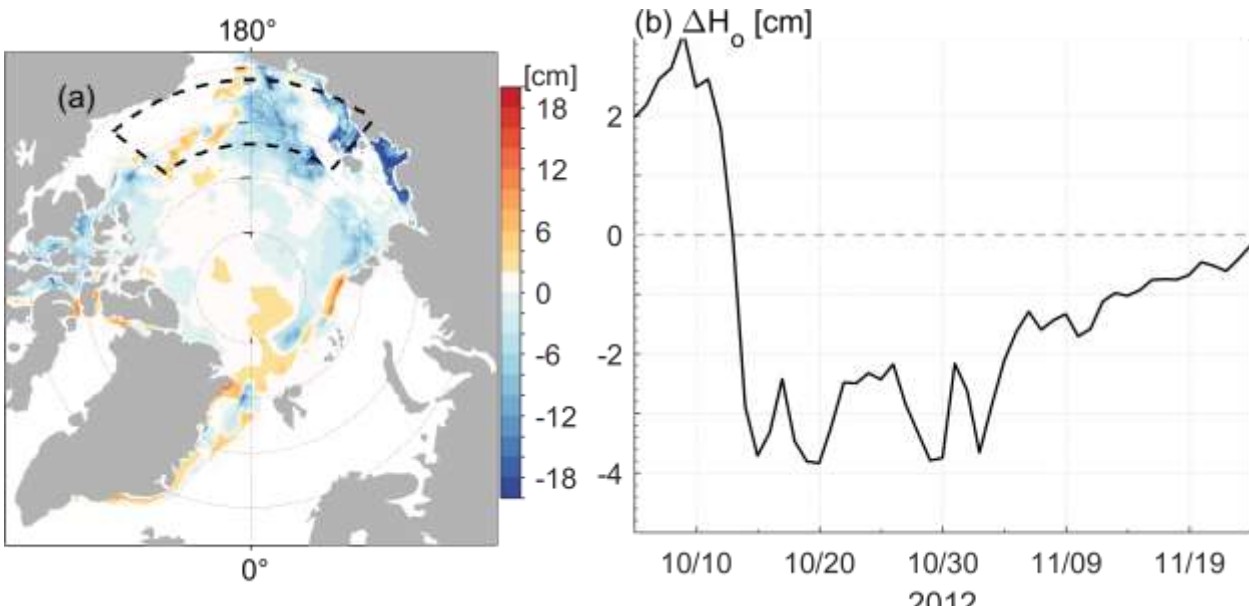

**Figure 6. (a) Adjustment on the parameter $H_o$ (in cm) averaged from October 12, 2012 to October 22, 2022. (b) Temporal**
**changes in adjustments of $H_o$ averaged over the Arctic's Pacific sector (enclosed by the dashed black line in (a)).**
**4.3 Comparisons against Independent Observations**
**4.3.1 Measurements at BGEP Moorings**
In this subsection, we interpolated the gridded SIC and SIT data to the BGEP mooring locations to explore their
seasonal variations.
The sea ice extent in 2012 reached the unprecedentedly low values since the satellite era. The ULSs of three
moorings observed ice-free conditions from August to October. The control run fails to simulate the ice-free conditions,
while both opti-SE and opti-SPE successfully reproduce the seasonal SIC variation (Figures 7a–c). opti-SPE shows
smaller RMSE at $M_b$ and $M_d$ than opti-SE and CTRL (see the numbers in Figure 7b, c). In opti-SPE, a pack ice broke
off from the main ice field from July 20 to early August and gradually melted at $M_a$, resulting in slightly larger RMSEs
(Figure 7a).
For SIT, the two assimilation runs bring the simulated SIT into consistency with the CS2SMOS observations
(Figure 7d–f) and ULS-measurements. From January to August, the gridded SIT (the CS2SMOS data and the two
assimilation runs) agrees well with the ULS-measured SIT, as the Beaufort Sea is covered by compact ice. However,
from October to December, when the ocean starts to freeze, systematic discrepancies emerge between the CS2SMOS,
the two assimilation runs, and the ULS-measurements: SIT increases more rapidly in the CS2SMOS dataset than in
either two assimilation runs or the ULS-measurements. Overall, the two assimilation runs exhibit the smallest errors
against the ULS-observed SIT and satellite-observed SIC, within the prior observational errors.

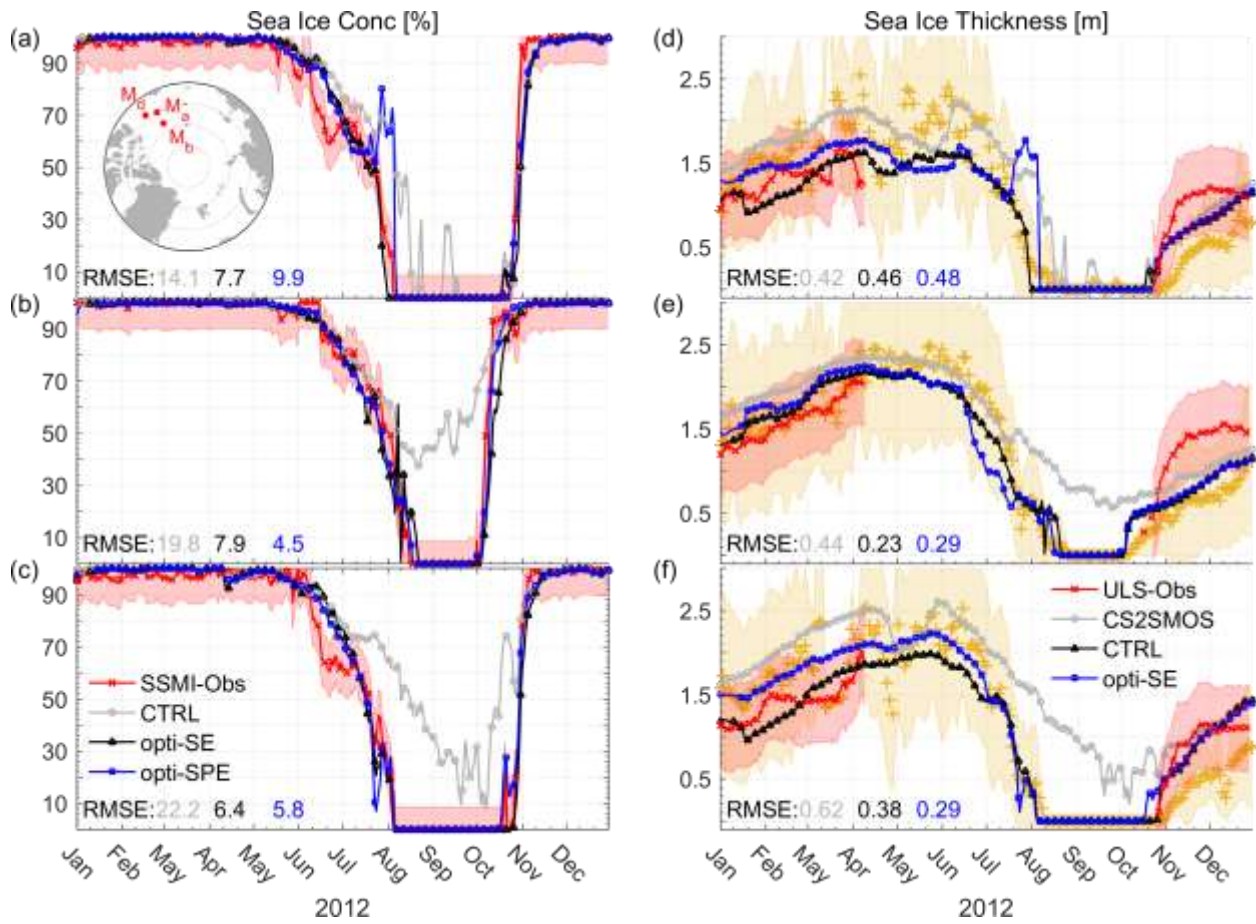

**Figure 7. SIC from satellite observations (SSMI-Obs), CTRL, opti-SE and opti-SPE at the mooring locations (a) Ma, (b) Mb, and (c) Md (see locations in subplot of panel (a)). The RMSEs of SIC between the three simulations and SSMI-Obs are listed in panels (a)–(c). Panels (d)-(f) are the corresponding SIT derived from the upward looking sonar (ULS-Obs), CS2SMOS, CTRL, opti-SE, and opti-SPE. And the RMSEs of SIT between the three simulations and ULS-Obs are also listed in panels (d)–(f). The shadings indicate the uncertainties of SIC and SIT obervations. For the ULS-Obs, we use the standard deviation of sub-daily sea ice thickness to represent the uncertainties.**

**4.3.2 SIT from IMB data**

The IMBs measure the variation of SIT along the drift trajectories. Comparing these Lagrangian SIT observations with gridded SIT from the sea ice-ocean coupled model and satellite data is challenging, as such comparisons are often subject to biases (e.g., Mu et al., 2018). We use two validation metrics: 1) the mean SIT over the observation time; 2) central root mean square deviation (CRMSD), computed as $\sqrt{\frac{1}{N}\sum_1^N\left((x_n - \overline{x_n}) - (y_n - \overline{y_n})\right)^2}$, where $x$ and $y$ represent the simulated- and measured-SIT, respectively, and the overbar denotes time average. $N$ denotes the total number of valid SIT observations from an IMB, and $n$ represents the index of individual observations. Smaller CRMSD values indicate better agreement between simulations and observations in capturing sea ice evolution. We selected six IMBs with long records: one deployed in 2011 in the central Beaufort Sea (2011J), which drifted toward the Chukchi Sea (top left panel in Figure 8) in 2012; the remaining five IMBs were deployed in 2012 in the central Arctic.

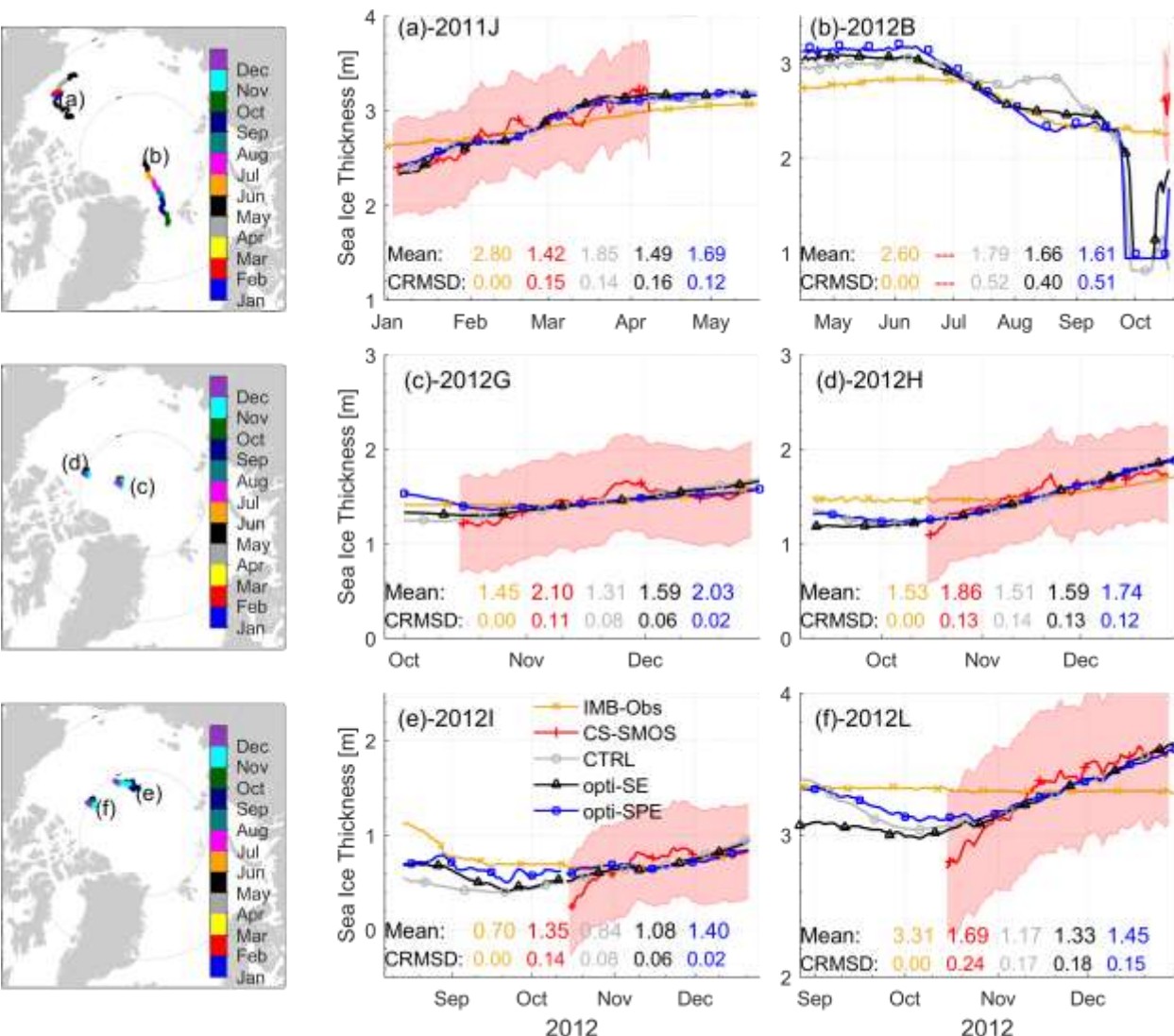

**Figure 8. SIT along the IMB trajectories. The IMB trajectories (a-2011J, b-2012B, c-2012G, d-2012H, e-2012I, f-2012L) are**
**shown in the left panels and the colors indicate the observing time. The SIT of IMBs (yellow lines with crosses), CS2SMOS**
**(red lines with plug signs), CTRL (grey lines with circles), opti-SE (black lines with triangles), opti-SPE (blue lines with**
**rectangles) are plotted in the right-hand side against observing time. The shadings indicate the CS2SMOS observational**
**uncertainties. The statistics (mean SIT and CRMSD against IMBs) of IMBs, CS2SMOS, CTRL, opti-SE, and opti-SPE are**
**also shown in each plot.**

As the mean SIT values indicate, SIT biases of 0.2−0.7 m exist between the CS2SMOS data and the control run
(see "Mean" values in Figure 8c−f). The optimization brings the model simulations closer to the CS2SMOS data,
especially in opti-SPE. When considering the SIT evolution along the drift trajectories, both opti-SPE and opti-SE
effectively reproduce the satellite-measured sea ice growth processes.
As expected, IMB measurements typically show 0.5−1.5 m differences compared with CS2SMOS observations
("Mean" values in Figure 8). This likely reflects that the ice conditions at the IMB deployment sites do not necessarily
represent a large spatial average, especially for SIT. Considering the SIT variation along the drift trajectories, all the
three model simulations and the IMB measurements exhibit much weaker SIT variability than the CS2SMOS
observations. This discrepancy arises from the following factors: 1) the buoys are generally deployed on the relatively
thick level ice (Richter-Menge et al., 2006), and 2) the satellite altimeter observations contain significant signals from
thin ice, which has a relatively large growth rate (Lei et al., 2022), and ice dynamic deformation can also contribute
to the thickening of sea ice within the grid cells (von Albedyll et al., 2022). The CRMSDs, computed between gridded
SIT and IMB measurements, indicate  that the SIT variations in opti-SPE matches the IMB measurement best.
In summary, opti-SPE effectively reduces the mean SIT biases between CS2SMOS observations and the control
run. The two assimilation runs reproduce the SIT variation observed by IMBs, with CS2SMOS observations exhibiting
greater variability than IMB measurements. Analysis of the CRMSDs reveals that opti-SPE best matches the sea ice
evolution as measured by IMBs. We note that data from point measurements of IMBs also have inherent limitations,
including the representativeness of initial ice thickness at the deployment, the influence of spatial heterogeneity of
snow accumulation during the later measurement stages, and the inability to account for sea ice thickening driven by
ice dynamic deformation.
**4.5 Sea Ice Velocity from IABP**
Twenty-two IABP ice-drifting buoys with drifting periods exceeding 100 days are used to retrieve ice velocities,
which are compared against the three model simulations. These buoys were initially deployed on multi-year sea ice
and subsequently drifted with the sea ice. Daily sea ice velocities are computed from the buoy locations, and the
model-simulated sea ice velocities are interpolated to the buoy locations. We summarize the statistics, including
standard deviations, correlation coefficients, and root mean square deviations (RMSDs), using Taylor diagrams
(Taylor, 2001).
The cost component of SID accounts for 1.45% of the cost function, decreasing slightly to 1.42% in opti-SE and
1.43% in opti-SPE. As shown in Figures 9b and 9c, the statistics of CTRL, opti-SE, and opti-SPE with respect to zonal
and meridional sea ice velocities from drifting buoys do not differ systematically. Based on the statistics, these buoys
can be categorized into two groups: 1) those buoys drifting in regions with compact ice (NOs. 1−14 in Figure 9a); and
2) those buoys drifting into regions with gradually decreasing SIC (NOs. 15−22 in Figure 9a).
In the first group (Nos. 1−14 in Figure 9a), the RMSEs (0.0429, 0.0417, 0.0423) and correlation coefficients
(0.8226, 0.8232, 0.8229) of CTRL, opti-SE, and opti-SPE are comparable. For individual buoys, their statistics in the
Taylor diagrams (Figure 9b, c) are also very similar across CTRL, opti-SE, and opti-SPE, with most RMSEs smaller
than the prior uncertainties (0.06 m s$^{-1}$).
In contrast to the first group, simulated ice velocities match the buoy observations (NOs. 15−22 in Figure 9a) much
less well in the second group, with RMSEs of 0.0880, 0.0973, and 0.0959, and correlation coefficients of 0.5691,
0.4863, and 0.4911 for CTRL, opti-SE, and opti-SPE, respectively. Additionally, the statistics of the second groups
in the Taylor diagrams are more dispersed, indicating that the two assimilation runs increase the sea ice velocity errors
and degrade correlations.

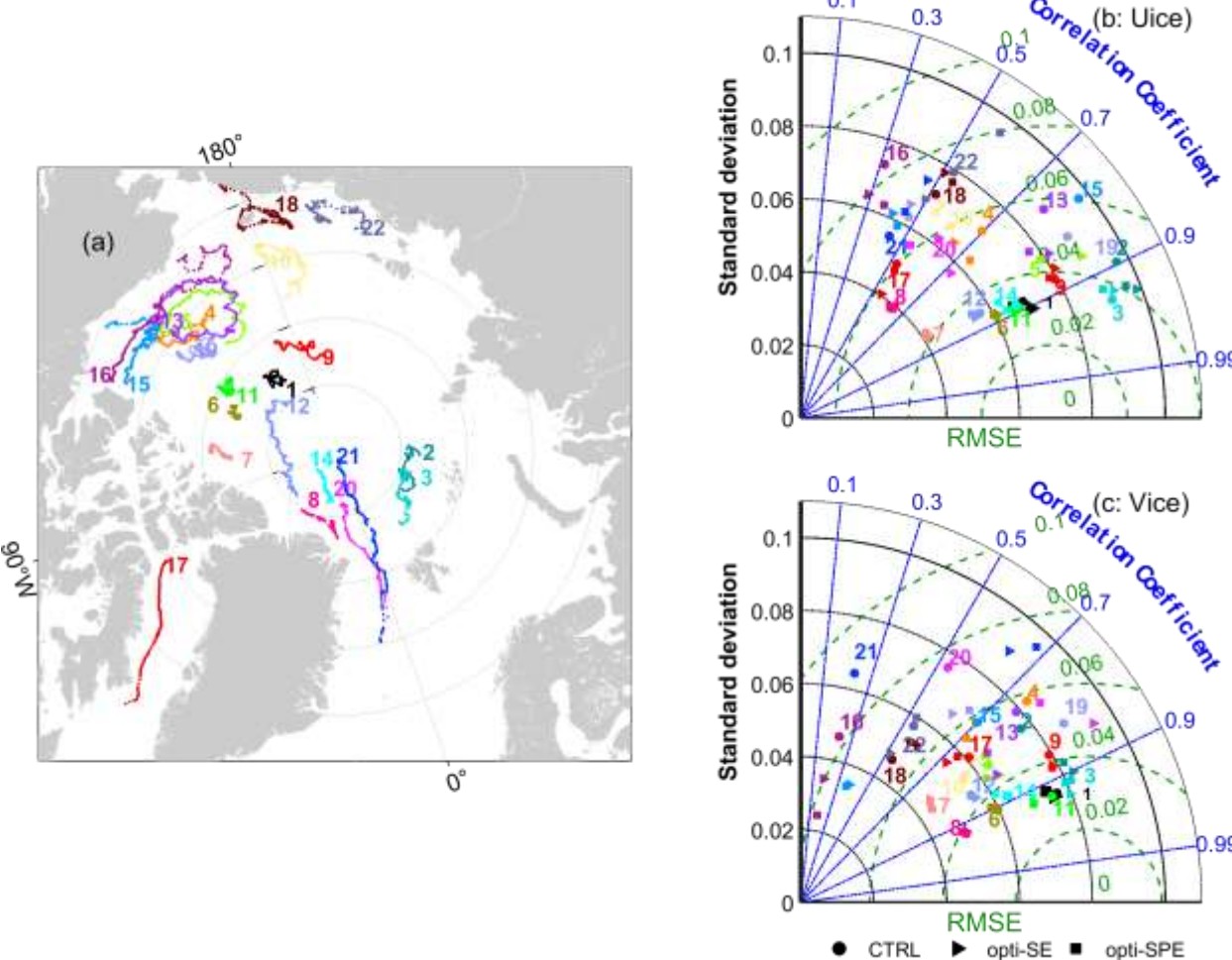

**Figure 9. (a) IABP buoy trajectories in 2012. Taylor diagrams of CTRL (filled circles), opti-SE (filled triangles), and opti-**
**SPE (filled squares) with respect to the (b) zonal (UICE) and (c) meridional (VICE) ice velocities retrieved from the IABP**
**buoys.**

In Figure 10, we further analyze two representative buoys (NOs. 16 and 21 in Figure 9a) that gradually drifted
toward the ice edge. The buoy-16 was deployed on April 12, 2012 in the southeastern Beaufort Sea (72.38 °N, -
127.47 °E). It drifted westward along the southern periphery of the Beaufort Gyre and ceased data reporting by
November 10, 2012 near the Alaska coast (71.34 °N, -160.55 °E). The buoy-21 was deployed on April 15, 2012 in the
central Arctic Ocean. It was advected along the transpolar stream, exited through the Fram Strait, and finally ceased
data reporting on November 6, 2012 in the Greenland Sea.
The sea ice velocity differences are very small in regions with compact ice, as ice internal stress dampens the
response of ice motion to wind forcing. When SIC decreases below 85% (Figure 10a, c), the marginal ice zone begins.
Ice internal stress weakens and plays a negligible role in sea ice drift. Consequently, sea ice becomes more sensitive
to wind stress (Figure 10b, d). In the marginal ice zone, the three model simulations cannot perfectly reproduce the
weakening ice field given current sea ice model physics, leading to noticeable errors in sea ice velocities. Along the
buoy-16's trajectory, the satellite-observed SIC approached zero by early August (Figure 10a), yet the buoy continued
drifting until late October (Figure 10b). The fragmented ice floes likely entered a fully free-drift state, where the ratio
of sea ice speed to wind speed reaches a maximum (e.g., Zhang et al., 2022). We suspect that the size of the ice floe
carrying the buoy was too small to be captured by the satellite and simulated by the numerical model which has a
resolution of ~10 km. This sea ice condition increases comparison uncertainty. Therefore, numerical simulation of sea
ice and characterization of heat and momentum exchange between the atmosphere and ocean are extremely
challenging in the marginal ice zone.

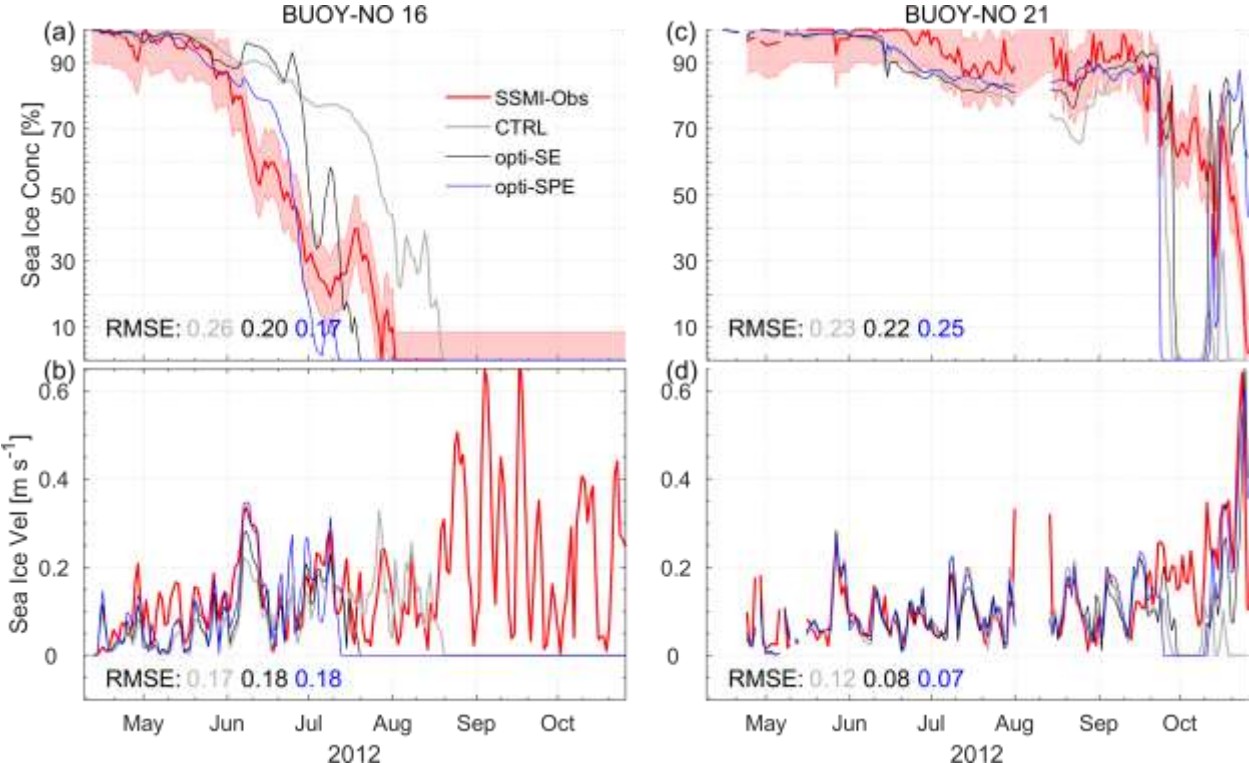

**Figure 10. Changes in (a, c) SIC and (b, d) sea ice velocity along the buoy trajectories (NOs. 16 and 21 in Figure 9a).**
**5 Conclusions and Discussion**
Building on the framework of the ECCO project, specifically the work of Fenty et al. (2013) and Lyu et al. (2023),
we developed a simultaneous state and parameter estimation scheme for an ocean-sea ice coupled model. To our
knowledge, this is the first study to optimize spatiotemporally varying sea ice model parameters together with
atmospheric forcing and model initial conditions. By assimilating Arctic ocean and sea ice observations from 2012,
we investigated the added effects of the simultaneous state and parameter estimation scheme (opti-SPE) relative to
the traditional state estimation scheme (opti-SE) and the free model simulation (CTRL).
The results show that opti-SPE further reduces the cost function by 10.5%, with notable improvements in SIC.
In both the control run and opti-SE, sea ice grows much slower in the Arctic's Pacific sector in October. By reducing
the lead closing parameter $H_o$ (on the order of 0.1 m), opti-SPE accelerates sea ice growth. The lead closing parameters
are widely used in Arctic sea ice-ocean assimilation models (Lindsay et al., 2006; Mu et al., 2018; Nguyen et al.,
2021), and impact on the sea ice growth process in autumn. Our results suggest that delicate optimization of this
parameter is necessary for better simulating Arctic sea ice changes and ocean-sea ice-atmosphere exchanges in the
autumn.
For SIT, both opti-SE and opti-SPE bring the simulated SIT into agreement with CS2CMOS SIT, taking into
account its prior uncertainties. Compared with the independent ULS-measured SIT at BGEP moorings, both
assimilation runs efficiently reproduce the seasonal sea ice evolution. The differences between the assimilation runs
and the ULS measurements are much smaller than those between CS2SMOS data and ULS measurements, especially
from October to December. Considering sea ice evolutions along the six IMB trajectories, opti-SPE performs slightly
better than opti-SE. However, the robustness of these comparisons may be limited by the small number of IMBs and
their spatial representativeness relative to their surrounding areas.
The overall improvements in sea ice velocities from data assimilation are not significant. In regions with compact
ice, the two assimilation runs slightly improve the simulated sea ice velocities. However, as buoys drift into the
marginal ice zone or near-ice-free regions, both model simulations and the satellite data may fail to simulate or detect
small ice floes, resulting in large deviations between the simulated and observed sea ice velocity.
Despite using a relatively low-complexity sea ice model compared to the modern sea ice models such as the Los
Alamos sea ice model (CICE; Hunke et al., 2020), our assimilation results demonstrate that the model simulations are
efficiently brought into agreement with the satellite observations and independent observations. Zampieri et al. (2021)
demonstrated that the low-complex sea ice model can be optimized more efficiently, and the overall performance after
optimization is largely in line with complex CICE configurations. Therefore, our sea ice model remains a suitable tool
for Arctic ocean and sea ice assimilation studies. However, in the next stage, we will update the sea ice module with
the more complex CICE model.
This study demonstrates that the simultaneous optimization of model parameters and state variables is promising
and merits testing with the other assimilation methods (e.g., ensemble Kalman Filters). However, given that the
optimal set of sea ice parameters may evolve alongside the thinning of Arctic sea ice, the parameters optimized using
2012 observations may not necessarily improve model simulation for the other years. To address this, we plan to
assimilate observations spanning the satellite era (1978−2025) and jointly optimize model parameters and state
variables. This approach will enable the accurate reconstruction of historical Arctic ocean and sea ice changes, thereby
supporting research on Arctic ocean and sea ice variability and trends.
**Competing interests**
The contact author has declared that none of the authors has any competing interests.
**Acknowledgments**
We thank Fran çois Massonnet from Universite catholique de Louvain and another anonymous for their constructive
and insightful comments on the manuscript. This research has been supported by the National Natural Science
Foundation of China (grant No. 42325604), the National Key Research and Development Program of China (grant
No. 2021YFC2803304), Program of Shanghai Academic/Technology Research Leader (Grant No. 22XD1403600),
and the Southern Marine Science and Engineering Guangdong Laboratory (Zhuhai) (Grant No. ML2023SP219). For
the assimilated sea ice variables, we thank the Alfred Wegener Institute, and EUMETSAT Ocean and Sea Ice Satellite
Application Facility for supplying the CS2SMOS L4 sea ice thickness, OSI-SAF sea ice concentration and sea ice
drift datasets, respectively. For the hydrographic observations, we acknowledge the Met Office, the Copernicus
Marine Service, and the remoting sensing system for archiving and sharing the EN4, the along-track SLA, and sea
surface temperature datasets. For the independent observations, we thank the Woods Hole Oceanographic Institution
for the sea ice draft measured by the ULS, the Cold Regions Research and Engineering Laboratory-Dartmouth Mass
Balance Buoy Program for the IMB data, the University of Washington for the buoy drifting trajectories.

**Author contribution**
Conceptualization: GL, AK, and LM. Experiments design and results analysis: GL. Investigation: RL, XL, and CL.
Original draft: GL. Review and editing: GL, LM, AK, RL, XL, and CL.

**Open Research**
**Code and Data Availability Statement**
The sea ice-ocean coupled model is based on MITgcm_c63m(https://mitgcm.org/download/other_checkpoints/). The
modified MITgcm (c63m) codes are available to the public on Zenodo at https://doi.org/10.5281/zenodo.14584929
(Lyu et al., 2025). The sea ice variables and model parameters from the control run and the two assimilation runs are
available on Zenodo at https://doi.org/10.5281/zenodo.14584780 (Lyu, 2025). We note that a commercial TAF license
is required to fully reproduce the optimization steps described in this study.
All the assimilated and independent validation are publicly available. The following datasets were provided with DOI
numbers: the global ocean along track L3 sea surface heights can be downloaded from the E.U. Copernicus Marine
Service Information (https://doi.org/10.48670/moi-00146); the sea ice concentration and drift data were based on the
EUMETSAT OSI-450-a (doi:10.15770/EUM_SAF_OSI_0013), and osi-405-c
(doi:10.15770/EUM_SAF_OSI_NRT_2007); the CryoSat-SMOS merged sea ice thickness data
(https://doi.org/10.57780/sm1-4f787c3, European Space Agency., 2023) is available at
https://spaces.awi.de/display/CS2SMOS/CryoSat-SMOS+Merged+Sea+Ice+Thickness.
The following datasets don't have DOI numbers, we provide their accessable links: the sea surface temperature were
retrieved from https://data.remss.com/SST/daily/mw_ir/v05.1/netcdf/ (last access: 13 April 2025, Gentemann et al.,
1994); the EN.4.2.2 data were obtained from https://www.metoffice.gov.uk/hadobs/en4/ (last access: 13 April 2025,
Good et al., 2013) and are ©British Crown Copyright, Met Office, 2013, provided under a Non-Commercial
Government Licence http://www.nationalarchives.gov.uk/doc/non-commercial-government-licence/version/2/ (last
access: 13 April 2025); the WOA18 climatological temperature and salinity data were download form
https://www.ncei.noaa.gov/access/world-ocean-atlas-2018/ (last access, 13 April 2025, Locarnini et al., 2018; Zweng
et al., 2018); the ULS-measured ice drafts were collected and shared by the Beaufort Gyre Exploration Program based
at the Woods Hole Oceanographic Institution (https://www2.whoi.edu/site/beaufortgyre/data/mooring-data/, last
access: 13 April 2025) in collaboration with researchers from Fisheries and Oceans Canada at the Institute of Ocean
Sciences; the IMB-measured sea ice thickness were from http://imb-crrel-dartmouth.org/results/ (Perovich et al., 2025,
last access: 13 April 2025); the sea ice drift trajecties were collected by the International Arctic Buoy Programme at
https://iabp.apl.uw.edu/data.html (last access: 13 April 2025).

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

*Levitus, edited by: Mishonov, A., NOAA Atlas NESDIS*, *82*, 50.