# Peer review of "Adjoint-Based Simultaneous State and Parameter Estimation in an Arctic Sea Ice-Ocean Model using MITgcm (c63m)"

_Geoscientific Model Development, 2024_

## Referee Comment (RC2)

Review of "Adjoint-Based Simultaneous State and Parameter Estimation in an Arctic Sea Ice-Ocean Model using MITgcm (c63m) »

by François Massonnet (UCLouvain). I did not look at the other reviewer's comment before submitting mine.

In this study, Lyu and colleagues employ the MITgcm and an adjoint method to jointly estimate the state and parameters of the ocean-sea ice model. They assimilate satellite and in-situ observations and compare the results of the CTRL run (no assimilation), adjoint-SE (state estimation only) and adjoint-SPE (state and parameter estimation) to independent observations. Most of the improvements come from the state estimation but further improvements are noted with the parameter estimation.

The study is interesting and the attempt to estimate parameters and state together has some future, I believe, especially in the context of prediction where model drift can be an issue.

My main remark on the paper is that the authors do the analysis on one year, and in fact on a very particular year : 2012. Why this choice ? Is there a risk that the outcome of the paper could be drastically different for other years? I am asking because 2012 is such a special year, with a strong cyclone (not something that is model-dependent) that may bias the results.

Related to that, I am not sure what is the overall implication of this work. Are the authors willing to recommend new parameter values for the MITGcm community? If so, I would like that they test whether the state is improved on a year without assimilation (e.g., running 2017 from the normal initial conditions but using parameters obtained for the 2012 estimation) and show the improvement.

The English could be improved at places, as suggested below.

- Line 18 : tunned  $\rightarrow$  tuned (many other instances in the text)
- Line 19 : of AN Arctic
- Line 27 : applied to perform/produce (not reproduce)?
- Line 33 : not sure that the processes themselves undergo changes : processes remain (e.g. heat conduction, ice melting), but it is the state of the system (affected by these processes) that changes
- Line 34 : To me, parameters stemming from parameterizations cannot be measured by construction. Nature ignores what a parameterization is.
- Line 35 : is assumed, not are assumed

- Line 42 : I would use the past tense as in the previous sentence. In general, please keep consistency of the tenses.
- Line 44 : budgetS
- Line 48 : sensitive
- Line 60 : likely TO improve
- Line 112 : The authors justify that assuming B^(-2) to be diagonal is a consequence of the fact that they rely on the adjoint model to project the modeldata misfits on the control variables. First, I do not understand what the two sentences have to do with each other. Second, I am surprised to read that B is assumed to be diagonal. In general, one can assume the observation error R to be diagonal (i.e., uncorrelated observational errors) but for the background model state, this seems to be a very strong assumption! Indeed model background errors are certainly correlated. Can the authors provide justification for the diagonal nature of B?
- Line 119 : uncertainties are set to 20% ; please be more specific : I assume this is the standard deviation of the error distribution (assumed Gaussian) ?
- Table 1 : could the authors justify where the ranges of parameters come from ?
- Line 141 : please specify what « uncertainty » means here.
- Line 156-158 : why multiplying here ?
- Line 161 : this is not very clear, especially when it is said that SIT is SIT x SIC. I would use another symbol, maybe SIT\_floe for the in-situ and SIT for the effective. Also, I assume that the "gridded SIT" means the model SIT?
- Line 202: "Firstly, the parameter changes within the range of uncertainties have considerable impacts on the model simulation. » is presented as «a prerequisite » but it is not. Did the authors mean « a hypothesis » maybe ? But then I am confused by the sentence after that. Maybe they meant « requirement » ?
- Line 207 « perturb by 10% » could mean many things : is that the range, the standard deviation, with perturbation statistical model ?
- Line 291 : this ocean heat
- Line 295 : a much
- Line 296. Starting a sentence with « While, « is strange
- Fig. 8 the colormap is not adapted for colorblind people, can you please choose a colorblind-friendly one?

---

## Author Comment (AC1)

**Response to comments on "Adjoint-Based Simultaneous State and Parameter Estimation in an Arctic Sea Ice-Ocean Model using MITgcm (c63m)" by François Massonnet (UCLouvain)**

Lyu et al.

Review of "*Adjoint-Based Simultaneous State and Parameter Estimation in an Arctic Sea Ice-Ocean Model using MITgcm (c63m)* »
by François Massonnet (UCLouvain). I did not look at the other reviewer's comment before submitting mine.

In this study, Lyu and colleagues employ the MITgcm and an adjoint method to jointly estimate the state and parameters of the ocean-sea ice model. They assimilate satellite and in-situ observations and compare the results of the CTRL run (no assimilation), adjoint-SE (state estimation only) and adjoint-SPE (state and parameter estimation) to independent observations. Most of the improvements come from the state estimation but further improvements are noted with the parameter estimation.

The study is interesting and the attempt to estimate parameters and state together has some future, I believe, especially in the context of prediction where model drift can be an issue.

My main remark on the paper is that the authors do the analysis on one year, and in fact on a very particular year : 2012. **Why this choice ? Is there a risk that the outcome of the paper could be drastically different for other years?** I am asking because 2012 is such a special year, with a strong cyclone (not something that is model-dependent) that may bias the results. Related to that, **I am not sure what is the overall implication of this work**. Are the authors willing to recommend new parameter values for the MITGcm community? If so, I would like that they test whether the state is improved on a year without assimilation (e.g., running 2017 from the normal initial conditions but using parameters obtained for the 2012 estimation) and show the improvement.

Response:

We thank François Massonnet for his valuable comments on our manuscript and for insightful discussions on results.

The second reviewer also raised concerns about the one-year assimilation window and the robustness of conclusion if applied to another year or period. As addressed in our response to the first reviewer, we selected the year 2012 as our test year with two reasons. First, the seasonal evolution of SIC errors in 2012 represent the general SIC errors pattern in the other years and the other Arctic ocean and sea ice reanalysis datasets. As shown in Figure R1 and Figure 4 (in the manuscript), the SIC errors increase to a peak in June, decrease slowly through September, and then rise sharply in October. A one-year assimilation window is sufficient to test whether joint optimization of parameters and state can reduce such seasonal SIC errors. Second, the 2012 sea ice retreat has been extensively investigated, providing a well-characterized reference to validate our assimilation results.

[Figure]

Figure R1. Root mean square errors of sea ice concentration between satellite observations and INTAROS-opt (black line, Lyu et al.,2021), TOPAZ4b (green line), and PIOMAS (magenta line).

In theory, when applying the data assimilation system to another year, data assimilation will adjust the control variables to bring the model simulations close to the observations for that specific year. The performance of such an application would depend largely on the quantity and quality of available observations of the target year.

Regarding the boarder implication, our work is not intended to propose new parameter values, as is common in parameter studies. Instead, we intend to developed an improved Arctic ocean-sea ice assimilation and reanalysis system and produce a new Arctic ocean and sea ice reanalysis dataset of higher accuracy. As demonstrated in our previous work (Lyu et al., 2021b, see the manuscript), our results are already comparable to the TOPAZ4 results and the others. With the simultaneous state and parameter estimation scheme presented in this study, we anticipate further improvements in the Arctic ocean-sea ice reanalysis dataset. We also suggest the other Arctic ocean-sea ice reanalysis system to try optimizing the model parameters, given that most existing Arctic ocean-sea ice reanalysis datasets exhibit similar SIC error patterns to those in our dataset.

The English could be improved at places, as suggested below.
We thank François Massonnet for his patience of correcting the English, we response the minor corrections one by one.

[1] Line 18 : tunned→tuned (many other instances in the text)
Response: Agree, we have revised these typos throughout the text.

[2] Line 19 : of AN Arctic
Response: Agree, we have added "an" in front of Arctic.

[3] Line 27 : applied to perform/produce (not reproduce) ?
Response: Agree, we have changed "reproduce" to 'produce'.

[4] Line 33 : not sure that the processes themselves undergo changes : processes remain (e.g. heat conduction, ice melting), but it is the state of the system (affected by these processes) that changes
Response: we thank the reviewer's comment, we revised "key processes" to "the ocean and sea ice state undergoes rapid changes"

[5] Line 34 : To me, parameters stemming from parameterizations cannot be measured by construction. Nature ignores what a parameterization is.
Response: That's true that nature ignores what a parameterization is. Likely, parameterizations are created by scientists. And therefore, we stated here, most of the parameters cannot measured. However, as far as I know, there are parameters which we can measure, such as ice and ocean albedo. Am I right?

[6] Line 35 : is assumed, not are assumed
Response: agree, and we have revised it.

[7] Line 42 : I would use the past tense as in the previous sentence. In general, please keep consistency of the tenses.
Response: agree, we have revised it.

[7] Line 44 : budgetS
Response: Agree, we have changed "budget" to "budgets".

[8] Line 48 : sensitive
Response: Agree, and has been corrected in Line 51.

[9] Line 60 : likely TO improve
Response: Agree, and we have corrected it.

[10] Line 112 : The authors justify that assuming $B^{-2}$ to be diagonal is a consequence of the fact that they rely on the adjoint model to project the model-data misfits on the control variables. First, I do not understand what the two sentences have to do with each other. Second, I am surprised to read that B is assumed to be diagonal. In general, one can assume the observation error R to be diagonal (i.e., uncorrelated observational errors) but for the background model state, this seems to be a very strong assumption! Indeed model background errors are certainly correlated. Can the authors provide justification for the diagonal nature of B?
Response:

There is no consequence between using a diagonal B and adjoint model, we deleted the word "therefore" in Line 123.

Definitely, the model background errors are correlated. In EnKF and 3DVar with assimilation window of days, we rely on these background correlations to achieve a mult-variate adjustments on the initial conditions. The background correlations are mostly statistical covariance or simplified model dynamic equations. In our system, both adjoint model and the background correlation can project the model-data mistfit to the control variable, realizing mult-variate adjustments. We use a one-year assimilation window in this study, which is long enough for the adjoint model to propagate the model-data misfits to all the model state and along the circulation pattern, and we prefer propagating these signal with model dynamics, rather than the statistical covariance in the background terms. Also, impacts of the initial conditions can hardly last for more than couple of months. Therefore, we choose to use a diagonal B, rather than a statistical covariance matrix or a diffusion model.

[11] Line 119 : uncertainties are set to 20% ; please be more specific : I assume this is the standard deviation of the error distribution (assumed Gaussian) ?
Response: That's true, we assume Gaussian error distribution with standard deviations of 20% of the corresponding parameter values. We revised it in lines 130-131.

[12] Table 1 : could the authors justify where the ranges of parameters come from ?
Response: some of the parameters ranges are from literatures (e.g., Sumata et al., 2019), such as albedo. I guess, they likely get the parameters ranges from modelling experts.

[13] Line 141 : please specify what « uncertainty » means here.
Response: we thank the reviewer for pointing out this problem. we mean observational errors here, rather than uncertainty.

[14] Line 156-158 : why multiplying here ?
Response: the SIC dataset doesn't provide observational error estimate, wen have to provide observational error by ourselves. The basic criteria is that: Considering the dependence of SIC product error on absolute value of SIC (Chen et al., 2023) and larger SIC errors off the coast due to the poor accuracy in the SIC product and the poor representation of landfast ice in the model.
   We choose this step function depending on SIC and multiplying these factors on the background errors. There are also other ways of defining the SIC errors.

[15] Line 161 : this is not very clear, especially when it is said that SIT is SIT x SIC. I would use another symbol, maybe SIT_floe for the in-situ and SIT for the effective. Also, I assume that the "gridded SIT" means the model SIT?
Response: we appreciated the reviewer's advice here. in our manuscript, SIT represent sea ice thickness averaged over the ice-covered region. Effective SIT (we changed it to grid-mean SIT) is the mean SIT over the model grid. Multiplying the grid area, we get the ice volume. We have change effective SIT to grid-mean SIT in Line 186.

[16] Line 202: *"Firstly, the parameter changes within the range of uncertainties have considerable impacts on the model simulation. »* is presented as «a prerequisite » but it is not. Did the authors mean « a hypothesis » maybe ? But then I am confused by the sentence after that. Maybe they meant « requirement » ?

Response: That's a good comment. Definitely this is not a hypothesis. The word "requirement" is better. if the two situations are met, the assimilation system si able to optimize the parameters. we have changed the word "prerequisites" to "requirements" in Line 202.

[17] Line 207 « perturb by 10% » could mean many things : is that the range, the standard deviation, with perturbation statistical model ?

Response: not that complex, we change the 13 model parameters (see Table 1) one by one by adding 10% of their default values. Overall, we run the forward model 13 times.

[18] Line 291 : this ocean heat

Response: Agree, we have corrected it.

[19] Line 295 : a much

Response: agree, we have corrected it.

[20] Line 296. Starting a sentence with « While, « is strange

Response: Agree, we have changed While to "And".

[21] Fig. 8 the colormap is not adapted for colorblind people, can you please choose a colorblind-friendly one?

Response: we thank the reviewer's advice. We have selected a colorblind-friendly colormap to show trajectory of the IMBs.

---

## Author Comment (AC3)

**Response to the reviewers' comments on "Adjoint-Based Simultaneous State and Parameter Estimation in an Arctic Sea Ice-Ocean Model using MITgcm (c63m)"**

Guokun Lyu et al.

This study potentially contributes significantly to the sea ice model optimisation problem by employing an adjoint method that, for the first time, simultaneously optimises both the model state and a set of its parameters. Therefore, the study topic is valuable and suitable for GMD, so its publication after a careful revision is recommended.

We thank the reviewers for their careful reading of the manuscript, their corrections of typos and grammatical issues, and their valuable suggestions for improving the work. We have revised the manuscript in accordance with the reviewers' comments. Below, we respond to the reviewers' questions and suggestions, with key points highlighted in red.

[1] A major issue is that the study is based on the very exceptional year 2011/2012 in the Arctic. However, the authors make general claims about the physical importance of sea ice model parameters. A question arises: Is one year enough to reach such conclusions, and are the results robust when tested for other years?

Response:

There are two reasons why we chose the year 2012 as our test base. Firstly, the seasonal evolution of sea ice concentration (SIC) errors represents the general pattern of SIC errors in other years and across other Arctic ocean and sea ice reanalysis datasets. As shown in Figure R1 and Figure 4 (in the manuscript), SIC errors increase and reach a peak in June; they then decrease slowly until September, followed by a rapid increase in October. A one-year assimilation window is sufficient to test whether the joint optimization of parameters and state can reduce seasonal SIC errors. Secondly, the sea ice retreat processes in 2012 have been extensively studied, which facilitates the validation of our assimilation results.

The assimilation results reveal that optimizing sea ice parameters improves the sea ice growth process in October. The optimization reduces the leads closing parameter H0 , which decreases the amount of latent heat required for the initial formation of sea ice. This leads to faster formation of thinner sea ice over open water (Figure 5 and Lines 291–303). The delayed sea ice recovery process occurs not only in 2012 but also in other years and across other Arctic ocean-sea ice assimilation models, as evidenced by the drastic increase in SIC errors during October in Figure R1. Therefore, we emphasize the importance of optimizing the lead closing parameter $H_0$. Moreover, $H_0$ is widely used in state-of-the-art sea ice modules and ocean-sea ice assimilation systems. Our

results indicate that optimizing such parameters is necessary to improve the accuracy of model simulations and forecasts. We have also added a discussion in Lines 472–475 regarding broader research implications in sea ice forecast and reanalysis.

[Figure]

Figure R1. Root mean square errors of sea ice concentration between satellite observations and INTAROS-opt (black line, Lyu et al.,2021), TOPAZ4b (green line), and PIOMAS (magenta line).

[2] There is an issue of reproducibility, as the adjoint model is only available to the editors and reviewers for review via the submission system, and the study may not comply with the FAIR principles (https://www.nature.com/articles/sdata201618).

Response:

We thank the reviewer for their comment. Prior to the review and discussion process, we had discussed this issue with the chief editor several times.

1) In this study, **we only modified the MITgcm (63m) model which we have published here** (https://doi.org/10.5281/zenodo.14584780 (Lyu, 2025))! With these MITgcm codes and data, anyone can reproduce the final results in the manuscript. So, in this sense, we have already meet the "Code and Data Policy" of GMD.

2) For the adjoint model of MITgcm, we and the MITgcm-ECCO communities didn't try to write them manually. This is different with the other modelling systems with adjoint (such as ROMS, WRF, etc.). MITgcm has been adapted for use with the TAF (Transformation of Algorithms in Fortran) to perform ocean state estimation and assimilation. Therefore, if you want to do assimilation with MITgcm, you need a TAF license, since "we (the TAF users) are not allowed pass the right of usage to any third party".

Following the Code and Data Policy of GMD, **we state clearly that: "We note that a commercial TAF license is required to fully reproduce the optimization steps described in this study."** in the "Code and Data Availability Statement".

Below is the Code and Data Policy of GMD: "where the authors cannot, for reasons beyond their control, publicly archive part or all of the code and data associated with a paper, they must clearly state the restrictions. They must also provide confidential access to the code and data for the editor and reviewers in order to enable peer review. The arrangements for this access must not compromise the anonymity of the reviewers.

[3] The discussion section is very brief, and what is crucially missing from it is comparing the study results with others, which would importantly put them in the context of broader research. Such a comparison would also help assess the results' robustness and novelty. Also, English is poor in places, with many writing mistakes and misspelled words. There is a need to check English grammar.

Response:

We appreciate the reviewer's comments here. Comparing different model simulations and reanalysis datasets is always valuable. We have attempted to compare our initial version of the Arctic ocean-sea ice reanalysis with TOPAZ4, PIOMAS, and ECCO. Recently, we identified issues with the seasonal evolution of SIC errors (Figure R1). To address this problem, we developed a simultaneous state and parameter estimation scheme. Our results indicate that optimizing parameters alongside initial conditions improved the SIC growth process in the Pacific sector of the Arctic Ocean during October. Notably, all other Arctic ocean-sea ice reanalysis datasets exhibit similar issues. Therefore, our findings encourage other assimilation systems to incorporate parameter estimation schemes into their assimilation frameworks.

Since this paper focuses on developing and testing the method, we only conducted a one-year assimilation experiment and compared the results with our own baseline assimilation outputs. We plan to run the assimilation system over the period 1979–2024 to produce a new Arctic ocean-sea ice dataset. Subsequently, extensive comparisons will be performed with other global and Arctic ocean-sea ice reanalysis products (e.g., Uotila et al., 2019).

Moreover, we have checked the English grammar carefully and revised the Conclusion and Discussion part.

Finally, I have multiple minor points and editorial suggestions that I hope the authors will consider:

[4] line 18: tuned

Response: we have corrected the typos.

[5] line 28: The end of Abstract mentions new Arctic reanalysis, but the manuscript does not provide any substantial information on that. I suggest you add such information, or delete the mention from Abstract.

Response:

We appreciate this comment. Our purpose in developing this system is to produce a new Arctic ocean-sea ice reanalysis dataset. This also represents the overall implication of this study and the system. We believe it is better to retain these words here.

[5] Introduction is missing to cite the relevant-looking paper by Xiying Liu and Lujun Zhang "Study on Optimization of Sea Ice Concentration with Adjoint Method," Journal of Coastal Research 84(sp1), 44-50, (1 June 2018). https://doi.org/10.2112/SI84-006.1 as they optimised MITgcm initial conditions. I suggest you add the citation.

Response: We have added the reference in Line 86.

[6] In introduction machine learning studies are mentioned. A potentially relevant study by Nie Y, Li C, Vancoppenolle M, et al (2023) Sensitivity of NEMO4.0-SI3 model parameters on sea ice budgets in the Southern Ocean. Geoscientific Model Development 16:1395–1425. https://doi.org/10.5194/gmd-16-1395-2023, is missing. I suggest considering adding it.

Response: We have cited this reference in Line 48.

[7] line 72: What are 'intermediate coupled models'? Do you mean intermediate complexity. Seems a word is missing here.

Response: Yes, we mean "coupled climate models of intermediate complexity". We have revised the words in Lines 82-83.

[8] line 98: 'z-levels ranging', again a word seems missing here. Do you mean z-level thicknesses?

Response: We thank the reviewer for pointing out the writing error. We have revised it to: "We have 50 z-levels with layer thickness ranging from 10 m at the surface to 456 m in the deep ocean."

[9] lines 108-114: This paragraph is hard to grasp. Could you clarify it e.g. by adding a sketch or a diagram illustrating a simple example?

Response:

Equation (1) is a general form of the cost function in variational assimilation. The first term on the right-hand side is easily understood: it represents the square of the model-data difference normalized by observational uncertainties. For simplicity, and using the parameter $H_0$ as an example, the second term on the right can be written as: $\frac{\Delta H_0^2}{\sigma^2}$. $\Delta H_0$ is the increment of the parameter $H_0$. $\sigma$ denotes the prior uncertainty of $H_0$, which is set to $\sigma=0.2 \times H_0$. This term increases with larger $\Delta H_0$ and thus limits the magnitude of adjustments to the parameters. Additionally, since the number of observations is less than the number of control variables, this term helps provide background information to prevent the optimization process from being underdetermined. Therefore, we state "ensures that complete information on the control variables is available" in Line 118.

We have attempted to clarify Equation (1) in a simplified manner but have not yet found a better approach. We hope the reviewer will understand it with the above explanation.

[10] Table 1 lists the parameters selected for optimisation. Why these parameters were selected has not been explained.

Response:

The 13 sea ice parameters were chosen based on a previous study by Sumata et al. (2019) (which used a similar sea ice module) and our prior sensitivity experiments. We have added the following sentence: "Based on a previous study with a similar sea ice module (Sumata et al., 2019) and our sensitivity experiments, we identify these 13 sea ice parameters, which have a considerable impact on sea ice properties" (Lines 136–138).

[11] line 165 and possibly elsewhere: Numbers and units have a space between them. '48h' -> '48 h'.

Response: We thank the reviewer for pointing out the errors. We have revised them throughout the manuscript.

[12] Table 2. The first column lists observational variables. Would be useful to explain in the caption what these acronyms mean. They are in text but caption seems more appropriate. In general, table and figure captions are quite brief and do not explain the figure and table contents very well.

Response: We thank the reviewer for their suggestions. We have added explanations for these acronyms below Table 2. For other tables and figures (Figures 1, 2, and 5) containing acronyms, we have also included explanations.

[13] line 171: 'include' -> 'includes'

Response: Agree, we have corrected this mistake.

[14] line 182: Does 'long measurement' mean long measurement period? It is not sure whether the IMB deployments on thick ice ensure the spatial representativeness, if the surrounding ice is thinner.

Response: The "long measurement" should be "long measurement period", we have corrected it in Line 218.

There appears to be a misunderstanding regarding the statement: "The IMBs are deployed on thick and level ice floes to achieve a relatively long measurement period and ensure the spatial representativeness of observations." Our intended meaning is that if IMBs drift over an extended period and distance, they can measure sea ice thickness (SIT) across a larger area, thereby achieving better spatial representation. Based on the reviewer's comment, "spatial representativeness" refers to how effectively a point SIT observation can represent SIT in its surrounding areas. To clarify this misunderstanding, we have deleted the phrase "ensure the spatial representativeness of observations."

[15] line 184: perhaps delete word 'fully'.

Response: We have deleted the word "fully".

[16] line 219: 'parameters' -> 'parameter'

Response: Agree, and we have corrected this typo.

[17] line 219: 'H0and evaluate' -> 'H0 and evaluate'

Response: We agree and have inserted a space here.

[18] lines 221-222: Are equations (2) and (3) representing a tangent linear model?

Response:

Equations (2)–(3) are not the tangent linear model itself, but rather schematic equations of a Taylor expansion. Using these equations (2)–(3), we can evaluate the linearity and nonlinearity of the coupled ocean-sea ice system through three forward

model runs. Within equations (2)–(3), $\frac{\partial M}{\partial H}$ can be interpreted as a tangent linear model. For assimilation applications, we developed the tangent linear model and its adjoint based on the discrete forward model, which is a complex and time-consuming process.

[19] line 231: Why does the model not reproduce positive SIC changes well?

Response:

The key point of Figure 2 is that the tangent linear model (TLM) can represent the size and pattern of SIC errors caused by parameter uncertainties of considerable magnitude. It is unclear why the TLM does not show any positive SIC changes. Several reasons may account for the failure to reproduce the positive SIC changes. Firstly, estimating the linear component using Taylor expansion requires the perturbation to be very small (e.g., $10^{-16} \sim 10^{-6}$, depending on nonlinearity of the model and integration time), but 10% perturbations on $H_0$ are large. In this case, the estimated linear component may not be accurate. Secondly, the tangent linear approximation of the nonlinear model and modifications to the TLM likely simplify some processes.

[20] line 278: Could you justify why October is important for Arctic sea ice?

Response:

October marks the refreezing period in the Arctic Ocean. A delayed refreezing process leads to thinner sea ice in subsequent years, creating a trend of increasingly thinner ice. Concurrently, more oceanic heat is released into the atmosphere, resulting in prolonged impacts on Northern Hemisphere weather patterns.

We have added the following sentence: "Since the sea ice recovery process has significant impacts on ocean-ice-atmosphere fluxes and sea ice thickness in subsequent years, we now focus on the SIC improvements during October" (Lines 292–293).

[21] line 280: 'most prounced over the Arctic Ocean.' -> 'most pronounced.'

Response: Agree, we have corrected the typo.

[22] Figure 5: What are red and green lines in panels (a), (b) and (c). You should explain them in the caption. What is the year in (d). Is it 2012? You should either add it to the caption or figure labels.

Response:

We thank the reviewer for their comments here. The green lines in Figures 5(a)–(c) indicate the sea ice edge (15% SIC) in the control run, and the red lines represent the

corresponding sea ice edge from satellite observations (Figure 5a), opti-SE (Figure 5b), and opti-SPE (Figure 5c). Since these lines were not mentioned in the text, we have removed them from Figure 5. In panel (d), we have added "2012" to the x-axis label.

[23] line 291: 'and this ocean heat'

Response: Agree, we have corrected the mistake.

[24] line 293: 'less ocean heat is released'

Response: Agree, we have corrected the mistake.

[25] line 294: 'water areas freeze'

Response: Agree, we have corrected the mistake.

[26] line 296: 'improve the SIC evolution'

Response: Agree, we have corrected the mistake.

[27] line 298: Did you look at how much the ocean heat varies between the simulations?

Response: We have not checked the ocean heat differences among the three simulations. The issue is that, with a one-year assimilation, adjustments to the initial temperature have a much greater impact on ocean heat changes than on sea ice. Once we complete our new reanalysis, it will likely be more insightful to compare ocean-ice-atmosphere heat fluxes across different reanalysis datasets.

[28] Figure 6: (b) Add the year (2012) in question.

Response: We thank the reviewer for their comment and have added "2012" to the x-axis label.

[29] line 314: broke off

Response: We agree and have corrected it.

[30] line 319: ocean starts to freeze

Response: We agree and have revised it.

[31] Figure 7: Add the year (2012) in question.

Response: We have added the year 2012 to x-axis label.

[32] line 329: SIC and SIC observations.

Response: Here, it should be "SIC and SIT observations". The left panels display satellite-derived SIC at the three mooring locations, with the shadings representing satellite SIC uncertainties. The right panels show SIT and their corresponding uncertainties.

[33] line 335: explain symbols N and n in the formula.

Response: We have added the following statement in Lines 396–397: 'N is the total number of valid SIT observations from an IMB, and n represents the observation number.

[34] line 337-338: 'with long records'

Response: We agree and has corrected the error.

[35] line 339: Are these IMBs still functional?

Response: These IMBs were deployed over 14 years ago, and none of them are operational anymore. In fact, new IMBs are deployed annually by CRREL (http://imb-crrel-dartmouth.org/results/), AWI, our group (PRIC), and others.

[36] Figure 8: Wouldn't it make more sense to calculate the statistics (mean and CRMSD) against CS-SMOS instead of IMBs. You seem to treat CS-SMOS as a reference.

Response: We thank the reviewer for their comment. The CS2SMOS SIT data are assimilated into the model and thus are not independent observations, whereas IMBs are independent observations. We compare the simulations against CS2SMOS data to assess whether the new algorithm can bring the model simulations closer to the assimilated observations. The comparison against IMBs is intended to validate the three model simulations. Additionally, we cross-validate the CS2SMOS data because this dataset does not consist of direct SIT measurements and may have certain unknown issues.

[37] line 347: Where do the 0.2-0.7 m SIT biases appear? Not in Figure 8.

Response:

In Figures 8a–f, we have listed the mean SIT at the bottom of each panel (marked as "Mean"). For example, in Figure 8a, the mean bias between the control run and CS2SMOS data is $1.85-1.42 = 0.43$ m.

When directly plotting the evolution of SIT, there are consistent vertical shifts between different lines. In such cases, the only signal visible is these biases. If we

remove these biases—for example, by adding an offset to the model results and CS2SMOS data—we can further compare the sea ice growth processes along their drifting trajectories. This is precisely what we try to illustrate in Figure 8.

[38] line 348: It is not clear that opti-SPE is closer to CS2SMOS than other simulations.

Response:

Here, we mean that the "Mean" SIT (see the mean value at the bottom of each panel) in opti-SPE is closer to that of CS2SMOS in Figures 8c–f. opti-SPE has reduced the mean SIT biases in these cases.

We have listed the mean SIT and plotted the SIT without offsets for two purposes: 1) to examine whether the state and parameter estimation algorithm can push the model further closer to the assimilated observations, as indicated by the "Mean" SIT; 2) to examine the SIT evolution in IMBs, model simulations, and CS2SMOS data.

[39] line 351: '... IMB measurements usually show 0.5-1.5 m differences compared ...'

Response: We agree and we have corrected it.

[40] line 355: Although IMB buoys are deployed on thick level ice, pressure ridges increase the areal average thickness of drift ice. Would this also increase CS2SMOs SIT?

Response: That is definitely a potential cause, which we have included in Line 357: "Ice dynamic deformation can also contribute to the thickening of sea ice within the grid." However, due to the relative low resolution and over-smooth, CS2SMOS SIT may not measure the pressure ridge efficiently.

 [41] line 368: 'drifting periods longer than'

Response: Agree and we have corrected it.

[42] Figure 9: Are these trajectories from Jan-Dec 2012?

Response:

Not all trajectories cover the period from January to December 2012. From the IABP dataset, we selected GPS trajectories recorded in 2012 with a duration of more than 100 days. In polar regions, most in-situ equipment is deployed between August and September. Some of this equipment may malfunction due to harsh environmental conditions, while others may drift out of the Fram Strait along with the ice.

Consequently, year-round Arctic observations are always scarce and valuable, and we have made every effort to utilize these data fully.

[43] line 381: 'the drifting buoys do not differ systematically.'

Response: Agree, we have corrected it following the reviewer's suggestion.

[44] line 383: 'the region with gradually decreasing SIC'

Response: Agree, we have corrected following the reviewer's suggestion.

[45] line 392: buoys 16 and 21 do not show reduction from CTRL to opti-SPE. Choose the buoys that do so, instead.

Response: We thank the reviewer for their comment here. The statistics in Figures 9b and 9c already show that the three simulations match the buoys well, with several exceptions (NOs 15-22). We have attempted to identify why data assimilation reduces the model performance, and we use buoys 16 and 21 to explain the reasons for this poor performance. The problem is that when the sea ice moves toward the ice-free regions, the model simulations and the satellite fails to simulate/observe the ice floe on which the buoys are deployed. Then the ice velocity doesn't match.

[46] line 399: Common understanding is that the MIZ starts when SIC is around 85% and the internal ice stress becomes unimportant.

Response: We thank the reviewer for their comment and have incorporated this common understanding in Lines 469–470.

[47] line 401: '(Figure 10b-d)'

Response: We thank the reviewer for pointing out this issue, and we have revised it to (Figure 10b,d) in Line 441.

[48] line 402: I do not understand this claim related to reproduction of the weakening ice field. In contrast, ice physics simulation becomes easier when the internal ice stress is reduced and ice drift becomes free. The RMSE could increase only due to faster SIV which is also more variable.

Response: It is true that simulating ice physics becomes easier when internal ice stress is reduced and ice drift is unconstrained. However, the issue is that sea ice models struggle to simulate SIC and SIT in regions with small ice coverage (e,g, Figure 10b, d). If ice floes are too small to be detected by either model simulations or satellite sensors, the sea ice velocity derived from GPS trajectories will not align with model simulations or satellite data. This is why we state, "It is not possible for the three model

simulations to perfectly reproduce the weakening ice field given current sea ice model physics."

[49 ]lines 408-409: Note that you use continuum ice model that can not capture individual ice floes.

Response: That is correct. This represents the key challenge when comparing ice-based observations with model simulations, particularly near the ice edge.

[50] line 411: 'the marginal ice zone'.

Response: We agree and we have corrected it.

[51] Figure 10: Is buoy 16 on the left and 21 on the right? Add relevant information in the figure caption.

Response: We thank the reviewer for their suggestion. We have replotted Figure 10, with buoy information included in the title, and have added "2012" to the x-axis.

[52] lines 415-452: English is very bad in this section. I suggest revising and rewriting entirely.

Response: We thank the reviewer for their suggestion and have revised this part.

[53] line 416: 'In a coupled sea ice-ocean model' and 'schemes describe'

[54] line 417: 'parameters are major sources'

[55] lines 417-419: This aspect was not addressed in the study, so the last sentence of this paragraph should be removed, or moved to Introduction with proper literature citations added.

[56] line 420: 'estimate the optimal model state and parameter values'

[57] line 425: 'simultaneous'

[58] line 426: 'traditional'

Response: we have rewrite the part related to comments [53]-[58].

[59] line 429: Note that sea ice does not grow laterally. There could be frazil ice growth that once buoyant enough starts to float on sea surface. Is that what you mean?

Response: We thank the reviewer for their suggestion. In the numerical model, frazil ice growth is parameterized using $H_0$. Reducing $H_0$ increases frazil ice growth. We have deleted the word "lateral" in Line 475.

[60] line 433: 'the simulated SIT in agreement with CS2CMOS SIT taking into account its prior uncertainties.'

Response: We thank the reviewer for their suggestion and have revised this in Lines 477–478.

[61] line 438: 'However, '

Response: We agree and have corrected the typo.

[62] line 447: 'brought into agreement' and 'efficiently, and also with independent'

Response: Agree, and we have corrected the typo.

[63] line 448: 'Zampieri et al. (2021) demonstrated '

Response: Agree, and we have deleted "also" as the reviewer suggested.

[64] line 450: 'Therefore, sea ice model remains a suitable tool for the Arctic Ocean'

Response: Agree, and we have corrected it.

[65] lines 450-451: 'studies. However, we will update'

Response: Agree, and we have corrected it.

[66] line 451: Is it computationally significantly more expensive to run the more complex CICE? Have you considered this aspect?

Response: We have begun updating the sea ice module with the more complex CICE model. At the current stage, computational cost does not appear to be a major challenge. However, we require time to develop the adjoint model of CICE and verify its accuracy, as we did for the results presented in Figure 2.

[67] lines 451-452: 'next state to reconstruct'

Response: We thank the reviewer for pointing out this typo. The correct wording should be: "We will update the sea ice module with the more complex CICE model in the next

stage and reconstruct changes in the Arctic Ocean and sea ice using SPE.". This is the primary objective behind our development of the assimilation system.

[68] line 466: 'University of Washington'

Response: Agree, and we have corrected the typo.

---

## Author Response (AR2)

**Response to comments on "Adjoint-Based Simultaneous State and Parameter Estimation in an Arctic Sea Ice-Ocean Model using MITgcm (c63m)" by François Massonnet (UCLouvain)**

Lyu et al.

The only point where I remain skeptical is on the justification of the use of one year only (2012) in the experimental setup. This point was also raised by the other reviewer, and I do not find a convincing reason why the experiment has not been replicated on another year. Is it because of a lack of computational resources? A lack of verifying observational data? To me, it is key to show the readers that the results obtained from one year are portable on other years; and if that hypothesis cannot be tested, then the authors must state the reasons.

On the statement in their response "In theory, when applying the data assimilation system to another year, data assimilation will adjust the control variables to bring the model simulations close to the observations for that specific year", I agree that it is how it *should* work, but can we have the proof? Can one simulation be done with the parameters of 2012 on another year? Again, this is to ensure the transferrability of the results to other years that the setup was tuned for. And if it is not possible, then the authors should acknowledge it.

Response:

We thank the reviewer for his thorough comments.

The use of a one-year assimilation window is primarily due to constraints on computational resources and time.

In this study, we demonstrate that the joint optimization of spatiotemporally varying parameters, initial conditions, and atmospheric forcing is technically feasible—and that this approach further improves sea ice simulations. This finding is particularly relevant for Arctic ocean and sea ice reanalysis, given that the optimal set of sea ice parameters may evolve alongside the thinning of Arctic sea ice. Additionally, optimizing the parameters of coupled ocean-sea ice models also merits testing with other assimilation methods (e.g., the Ensemble Kalman Filter, EnKF).

As we anticipate, the optimized parameters may not necessarily improve simulations for other years, given that the optimal set of sea ice parameters is likely to evolve alongside the thinning of Arctic sea ice. To address this, we plan to assimilate observations spanning the satellite era (1978–2025) and jointly optimize model parameters and state variables. This approach will enable the accurate reconstruction of historical Arctic ocean and sea ice changes, thereby supporting research on Arctic ocean and sea ice variability and trends

To avoid over-interpretation of this study and clarify our purpose, we have added the following text:

(1) in the Abstract (L26-29 ): "Given that the optimal set of sea ice parameters may evolve alongside the thinning of Arctic sea ice, the adjoint-based SPE scheme has the potential to more accurately reconstruct the histical Arctic ocean and sea ice changes

covering the satellite era, supporting research on Arctic sea ice and ocean variability."

(2) in the Conclusion and Discussion (L455-461): "This study demonstrates that the simultaneous optimization of model parameters and state variables is promising and merits testing with the other assimilation methods (e.g., ensemble Kalman Filters). However, given that the optimal set of sea ice parameters may evolve alongside the thinning of Arctic sea ice, the parameters optimized using 2012 observations may not necessarily improve model simulation for the other years. To address this, we plan to assimilate observations spanning the satellite era (1978–2025) and jointly optimize model parameters and state variables. This approach will enable the accurate reconstruction of historical Arctic ocean and sea ice changes, thereby supporting research on Arctic ocean and sea ice variability and trends."